# Potential greenhouse gas reductions from Natural Climate Solutions in Oregon, USA

Rose A. Graves[1,2]*, Ryan D. Haugo[2], Andrés Holz[3], Max Nielsen-Pincus[4], Aaron Jones[2], Bryce Kellogg[2], Cathy Macdonald[2], Kenneth Popper[2], Michael Schindel[2]

1 College of Liberal Arts and Sciences, Portland State University, Portland, Oregon, United States of America, 2 The Nature Conservancy, Portland, Oregon, United States of America, 3 Department of Geography, Portland State University, Portland, Oregon, United States of America, 4 Department of Environmental Science and Management, Portland State University, Portland, Oregon, United States of America

* rose.graves@tnc.org

**Data Availability Statement:** All relevant data are within the manuscript and its Supporting Information files. In addition, all simulation R code and input data files are available from the lead

## Abstract

Increasing concentrations of greenhouse gases (GHGs) are causing global climate change and decreasing the stability of the climate system. Long-term solutions to climate change will require reduction in GHG emissions as well as the removal of large quantities of GHGs from the atmosphere. Natural climate solutions (NCS), i.e., changes in land management, ecosystem restoration, and avoided conversion of habitats, have substantial potential to meet global and national greenhouse gas (GHG) reduction targets and contribute to the global drawdown of GHGs. However, the relative role of NCS to contribute to GHG reduction at subnational scales is not well known. We examined the potential for 12 NCS activities on natural and working lands in Oregon, USA to reduce GHG emissions in the context of the state's climate mitigation goals. We evaluated three alternative scenarios wherein NCS implementation increased across the applicable private or public land base, depending on the activity, and estimated the annual GHG reduction in carbon dioxide equivalents ($CO_2$e) attributable to NCS from 2020 to 2050. We found that NCS within Oregon could contribute annual GHG emission reductions of 2.7 to 8.3 MMT $CO_2$e by 2035 and 2.9 to 9.8 MMT $CO_2$e by 2050. Changes in forest-based activities including deferred timber harvest, riparian reforestation, and replanting after wildfires contributed most to potential GHG reductions (76 to 94% of the overall annual reductions), followed by changes to agricultural management through no-till, cover crops, and nitrogen management (3 to 15% of overall annual reductions). GHG reduction benefits are relatively high per unit area for avoided conversion of forests (125–400 MT $CO_2$e ha$^{-1}$). However, the existing land use policy in Oregon limits the current geographic extent of active conversion of natural lands and thus, avoided conversions results in modest overall potential GHG reduction benefits (i.e., less than 5% of the overall annual reductions). Tidal wetland restoration, which has high per unit area carbon sequestration benefits (8.8 MT $CO_2$e ha$^{-1}$ yr$^{-1}$), also has limited possible geographic extent resulting in low potential (< 1%) of state-level GHG reduction contributions. However, co-benefits such as improved habitat and water quality delivered by restoration NCS pathways are substantial. Ultimately, reducing GHG emissions and increasing carbon sequestration to combat climate change will require actions across multiple sectors. We demonstrate that

author's GitHub repository (https://github.com/rosegraves/OregonNCS).

**Funding:** RAG was supported by grants from The Nature Conservancy (Natural Climate Solutions Fellowship Grant #2018-Climate Fellow-1), Portland State University Institute for Sustainable Solutions, and Portland State University College of Liberal Arts and Sciences Dean's Office. The funders had no role in study design, data collection and analysis, decision to publish, or preparation of the manuscript.

**Competing interests:** The authors have declared no competing interests exist.

the adoption of alternative land management practices on working lands and avoided conversion and restoration of native habitats can achieve meaningful state-level GHG reductions.

## Introduction

Limiting climate change and temperature increases to below 1.5 to 2˚C is critical to maintaining stability in human and environmental systems [1]. Stabilizing global climate will require rapid and targeted actions to reduce greenhouse gas (GHG) emissions. While fossil fuel mitigation and transitioning to clean energy systems will be required to combat climate change, most scientists now agree that additional activities will be needed to rapidly reduce GHG emissions and avoid the worst effects of climate change [1–4]. Natural climate solutions (NCS), or changes in land management, ecosystem restoration, and conservation on natural and working lands as part of GHG reduction strategies, can provide valuable co-benefits for people and nature while contributing to climate mitigation [5–8]. NCS provide climate benefits through two major mechanisms: (1) avoiding emissions by limiting conversion or altering management activities that lead to loss of stored carbon or increased GHG emissions; or (2) increasing carbon sequestration and storage through ecosystem restoration or altered land management regimes. Global- and national-scale NCS evaluations suggest that, if enacted rapidly (i.e. within the next 10 to 15 years), these activities could contribute up to 30% of the needed global near-term climate mitigation to limit warming to 2˚C [9] and offset the equivalent of 21% of current net GHG emissions in the United States [8].

The role of subnational governance, policy, and actions is increasingly important for combatting climate change. Subnational commitments to reduce GHG emissions have become more common in the past decade in response to faltering multinational agreements and lack of comprehensive national climate policy [10–12]. Some states are able to implement policies and facilitate GHG reductions where national governments have been unable to make progress [13,14]. For example, nine states in the northeastern and mid-Atlantic U.S. formed the Regional Greenhouse Gas Initiative (RGGI, www.rggi.org) to cap and reduce emissions from the energy sector [15] while the state of California successfully passed first-of-its-kind legislation mandating a state-wide cap on emissions [16]. More recently, an increasing number of states have pledged or legislated goals targeting net-zero emissions by 2050 [17]. Ambitious goals of zero or negative emissions will only be achievable by including the NCS potential [2]. States are often able to be more nimble or experimental with their policies, illustrating possible strategies which could be replicated at larger scales [14].

While global and national scale NCS evaluations provide a starting point for policy conversations, subnational decision-makers require information at a corresponding scale. A bi-partisan coalition of Governors have joined the U.S. Climate Alliance, and committed to reduce greenhouse gas emissions consistent with the goals of the Paris Agreement, including identifying best practices for land conservation, management and restoration in carbon policies [18]. Recently, California recognized the potential for land management to contribute to emission reductions [19], promoting assessments of the NCS potential on that state's natural and agricultural lands [20]. Using an approach that acknowledged uncertainty in the exact GHG reductions attributable to each NCS activity, Cameron and colleagues [20] found that NCS could contribute up to 17% of California's GHG reduction goals by 2030. While California consistently provides a leading example in state-level climate action [10], other states have

been slow to follow its lead, citing concerns over economic costs and political uncertainty over the need to limit GHGs [21]. As other member states within the US Climate Alliance grapple with how to aggressively reduce GHGs, NCS evaluations from additional states can help to refine the coarser scale global and national analyses and provide a range of options for state and non-state actors to consider when developing programs and policies to address climate change [11,22].

To address the need for applied NCS science at the subnational level, we adapted the framework presented by Cameron and colleagues (2017) to evaluate the potential contribution from NCS activities to GHG reduction goals in Oregon. Located in the U.S. Pacific Northwest, Oregon has a long history of strong land use controls and environmental policy [23–25] and initiated a task force on global warming over three decades ago in 1988 [26]. In 2007, the Oregon Legislature established GHG reduction goals setting a target for statewide emissions to be limited to 75% below 1990 levels, or 14 MMT $CO_2$e, by 2050 with an interim target of 33.9 MMT $CO_2$e by 2035 (HB 3543; www.keeporegoncool.org). In addition, Oregon recently joined the U.S. Climate Alliance and has committed to including natural and working lands in GHG emission reduction strategies. This study contributes to our evolving understanding of the potential for the land sector to mitigate climate change.

## Methods

### General analytical framework

We simulated the potential GHG reduction attributable to each of 12 NCS activities (Table 1) between the years 2020 and 2050 under three potential implementation scenarios. NCS activities were chosen based on applicability to natural and working lands within Oregon and their ability to directly achieve co-benefits for the conservation of biodiversity. Current rates of each activity were compiled from multiple data sources and served as the baseline for all scenarios (Table 2). We used empirical values from peer reviewed or government gray literature to develop estimates of the GHG emissions and/or carbon sequestration attributed to each activity (Fig 1). We then created three implementation scenarios wherein we modified the implementation rate of each activity in order to decrease emissions or increase carbon sequestration, relative to the baseline.

For each implementation scenario, we used Monte Carlo simulation to account for the uncertainty associated with carbon sequestration and GHG emission values. For each NCS activity and each simulation year, we sampled 1000 iterations from a distribution created from the uncertainty range for that activity. Specific distributions and details on the calculation of the associated reductions and implementation scenarios are described below. We report the range of possible GHG reductions in $CO_2$e from each NCS activity and provide estimates of the uncertainty surrounding each of those estimates. We then compare the reduction potential of the activities against Oregon's GHG reduction targets to highlight the contributions of these activities. All simulations and analyses were conducted using R (version 3.4.1).

### Avoided conversion NCS

**Forests to development.** We estimated the current rate of forest conversion on private land in Oregon using published land use data and standardized statewide land use maps from 1994 to 2014 [27]. We calculated the annual conversion rate of forests to urban and to low-density residential or agricultural (i.e., rural) land uses by county.

To quantify emissions from forest conversion, we estimated pre-conversion carbon stocks using the USFS Forest Inventory Analysis (FIA) Evalidator application (https://apps.fs.usda.gov/Evalidator/evalidator.jsp). We extracted forest carbon (t C ha$^{-1}$) in each of the IPCC

**Table 1. Descriptions of the activities included in the natural climate solutions pathway analysis for Oregon's natural and working lands.** NCS activities and their definitions are adapted from Cameron et al. [20], Griscom et al. [7], and Fargione et al. [8].

| | Natural Climate Solution Activity | Description |
|---|---|---|
| **Avoided Conversion** | Avoided conversion of forests to rural development | Emissions avoided by limiting anthropogenic conversion of forests to low-density and agricultural development |
| | Avoided conversion of forests to urban development | Emissions avoided by limiting anthropogenic conversion of forests to high-density, urban development |
| | Avoided conversion of sagebrush-steppe to invasive annual grasses | Emissions avoided by limiting the conversion, post-fire, of sagebrush-steppe to invasive annual grasses; assumes active management of sagebrush-steppe recovery |
| | Avoided conversion of grasslands to tilled cropland | Emissions avoided by limiting the anthropogenic conversion (e.g., tilling) of existing grassland to intensive agriculture. |
| **Land Management** | Deferred timber harvest | Avoided emissions and increased sequestration associated with deferring harvest on a portion of Oregon's forest. We consider timber harvest across all forest ownerships in Oregon, but limit deferred harvest to counties with lower risk of wildfire (e.g., western Oregon). |
| | Use of cover crops | Increased carbon sequestration due to use of cover crops, either to replace fallow periods between main crops or as inter-row cover in specialty crops such as orchards, berries, and hops. |
| | No-till agriculture | Increased carbon sequestration due to the use of no-till agriculture on tilled cropland. |
| | Nutrient management | Avoided emissions by improving N fertilizer management on croplands, through reducing whole-field application or through variable rate application. |
| **Restoration** | Replanting after wildfire on federal land | Increased carbon sequestration from increased post-wildfire reforestation on managed federal lands (e.g., wilderness areas are not included). This NCS assumes no salvage harvest or site-prep before replanting. |
| | Riparian forest restoration | Increased carbon sequestration through active replanting of forest along non-forested riparian areas. |
| | Tidal wetland restoration | Increased carbon sequestration due to restoring tidal processes where tidal wetlands were the historical natural ecosystem; limited to the most highly saline historical tidal wetlands. |
| | Invasive annual grasses to sagebrush-steppe | Increased carbon sequestration due to restoring sagebrush-steppe ecosystems in areas dominated by invasive annual grasses. |

carbon pools (i.e., above-ground biomass *AGB*, belowground biomass *BGB*, litter *L*, woody debris *WD*, and soil organic matter) for private lands in each county [28]. To quantify uncertainty in pre-conversion carbon stocks, we grouped counties across the interior versus coastal PNW productivity gradient simplified to east and west of the Cascade Mountains (S1 Table). Using the estimates, sampling errors, and number of plots reported by Evalidator, we calculated the pooled mean and pooled standard deviation for the interior and coastal regions. For Monte Carlo simulations, we randomly sampled from normal distributions constructed with the pooled mean and standard deviations. We assume only partial emission of live above-ground carbon stocks, using an emissions factor (EF) of 54% following Fargione et al. [8] which reflects the fact that some harvested carbon will be retained in wood products and other harvest processes. For conversion to urban development, we assume a complete conversion of the belowground biomass, litter, and woody debris pools whereas we assume partial conversion of these pools for rural development (50%). We do not include emissions from soil

**Table 2. Current annual implementation rates for activities included in the natural climate solutions pathways analysis for Oregon's natural and working lands.**

|  | Activity | Baseline (Current Annual Rate) |
|---|---|---|
| **Conversion** | Forests to rural development | 1930 ha [a] |
|  | Forests to urban development | 148 ha [a] |
|  | Sagebrush-steppe to invasive annual grasses | 4000 ha [b] |
|  | Grassland to agriculture | 930 ha [c] |
| **Land Management** | Timber harvest | 3.4 billion board feet [d] |
|  | Cover crops | 48,740 ha [e] |
|  | No-till agriculture | 403,280 ha [e] |
|  | Nutrient management | 193,000 Mg N [f] |
| **Restoration** | Replanting after wildfire on federal land | 9–12% of moderate to high-severity burned area [g] |
|  | Riparian forest restoration | 2395 ha [h] |
|  | Tidal wetland restoration | 49 ha [h] |
|  | Invasive annual grasses to sagebrush-steppe | 5590 ha [b] |

Historical data range

[a] 1994–2014

[b] 2009–2014

[c] Land use change from 2008–2012

[d] Harvest data by ownership 2000–2017 for counties with less than 50% of forest area at high risk of wildfire (S2 Fig)

[e] 2012 and 2017

[f] 1997–2017

[g] 2000–2015

[h] 1998–2017

organic matter, as the overall effects of conversion from forests to residential development on soil carbon are unclear [29,30]. Finally, we converted from t C to $CO_2$e using a conversion factor of 44/12 for $CO_2$e to carbon. Thus, committed emissions are equal to:

$$Forests\ to\ Urban = (AGBx0.54 + BGB + L + WD)\ X\ \frac{44}{12}$$

$$Forests\ to\ Rural = (AGBx0.54 + BGB + L + WD)X\ 0.5\ X\ \frac{44}{12}$$

In addition to the initial loss of carbon stocks, we estimated the ongoing carbon sequestration that would be lost due to forest conversion. Using the USFS FIA Evalidator, we extracted the gross annual growth for private lands by county and then grouped counties into the interior or coastal region [31]. As above, we quantified uncertainty in pre-conversion sequestration by calculating the regional pooled mean and standard deviation from the county-level estimates, sampling errors, and number of plots reported by Evalidator. For Monte Carlo simulations, we randomly sampled from a normal distribution constructed from the pooled mean and standard deviation. We converted reported gross annual growth ($ft^3$ $acre^{-1}$) to $MTCO_2$e $ha^{-1}$ using specific gravity estimates from Smith et al. [32]. We assumed that conversion of forests to urban development resulted in a loss of 84% of forest carbon sequestration [33], while forests to rural development resulted in 50% of loss of forest carbon sequestration [34].

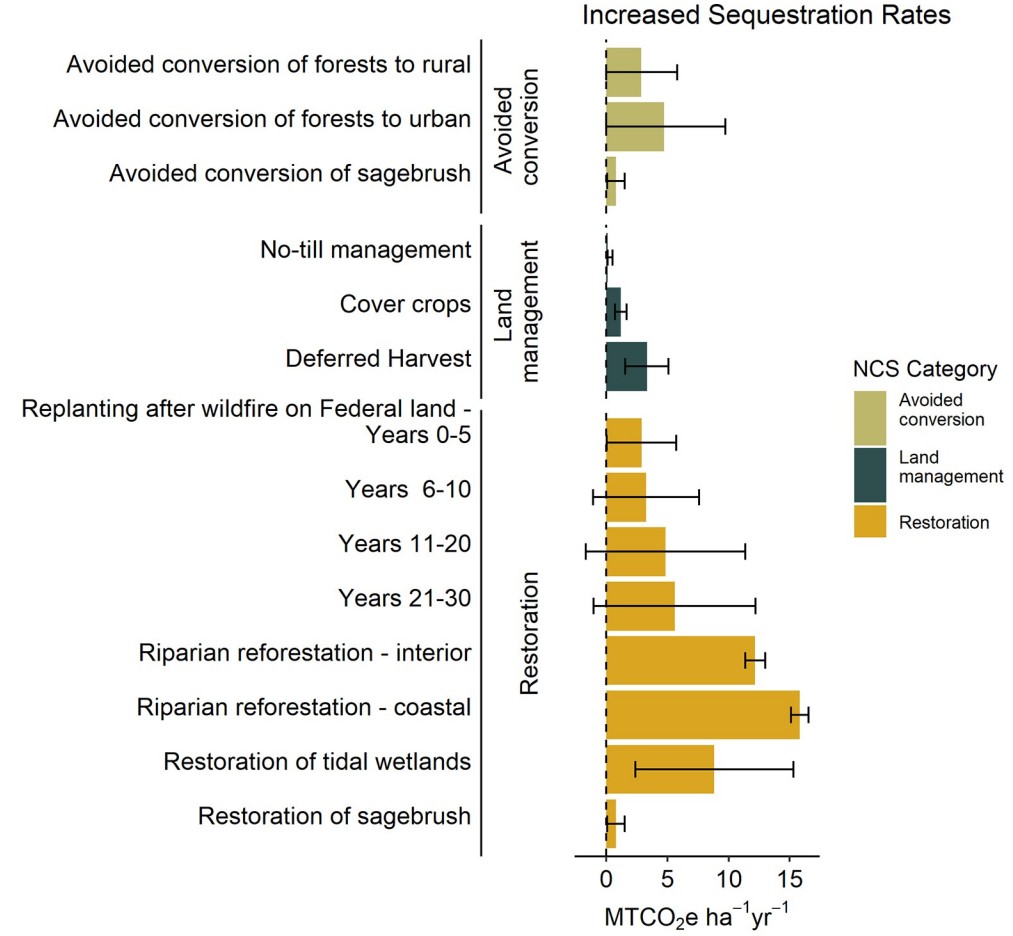

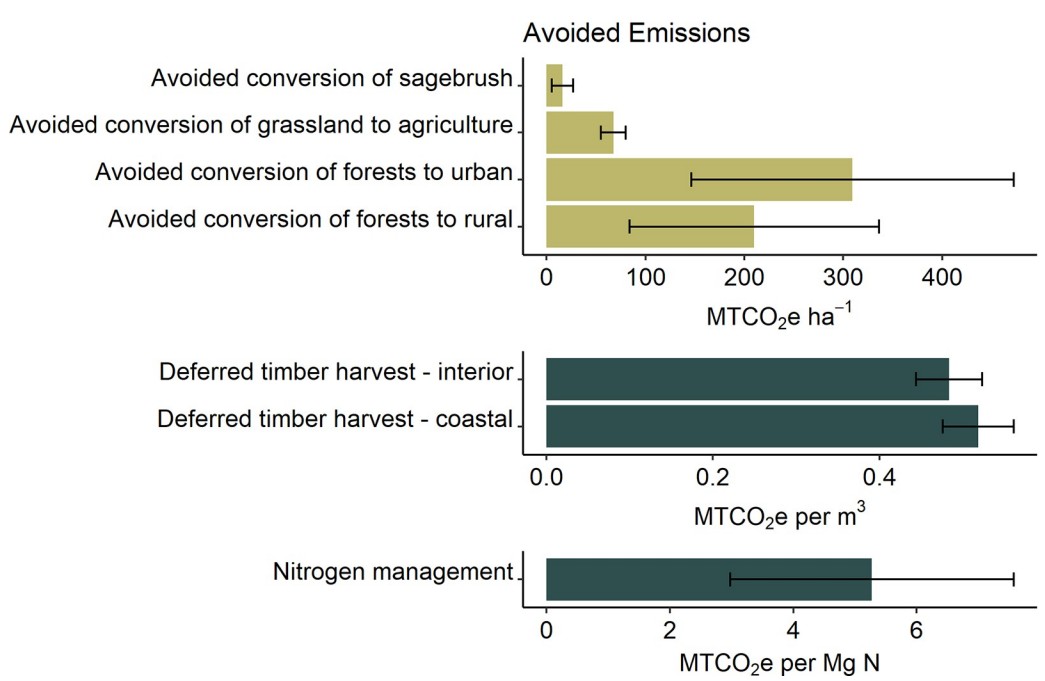

**Fig 1.** Values used to parameterize the Monte Carlo simulations for (A) increased sequestration and (B) avoided emissions. Some activities have varying rates of sequestration or avoided emissions depending on their location relative to the interior vs. coastal productivity gradient or based on forest age. Error bars represent the 90% confidence interval.

**Sagebrush-steppe to invasive annual grasses.** Conversion of sagebrush-steppe to invasive annual grasses results in a one-time loss of stored carbon as well as an ongoing loss in carbon sequestration [35]. Wildfire increases the likelihood of invasion by annual grasses, especially in the relatively warm and xeric portions of the northern Great Basin [36]. To estimate current rates of sagebrush-steppe conversion to invasive annual grasses, we combined data on areal extent of fires in the region and annual grass dominance using the Burned Areas Boundaries Dataset 1984–2014 [37] and the Estimated Ecological States dataset [38]. From 1984 to 2014, the mean area burned was 40,000 ha yr$^{-1}$. We calculated the background level of invasion by annual grasses as the proportion of invasive annual grass dominated land outside of burned areas (13%) and subtracted that from the proportion of burned areas dominated by invasive annual grasses (23%) to estimate a conversion rate of 10% of burned areas, or 4000 ha yr$^{-1}$.

We used published estimates of aboveground biomass loss and changes in carbon sequestration due to conversion of sagebrush-steppe to invasive annual grasses [39]. Estimates of aboveground biomass carbon loss ranged from 4.03 to 23.83 MTCO$_2$e ha$^{-1}$, with a mean and standard deviation of 16.13 ± 6.6 MTCO$_2$e ha$^{-1}$. We do not include belowground biomass loss, because the effect of sagebrush conversion on belowground carbon storage is highly uncertain [39–41]. Estimates of post-fire invasive annual grass carbon sequestration are significantly lower than average sagebrush carbon sequestration [42] and are summarized as foregone carbon sequestration of 0.81 ± 0.44 MTCO$_2$e ha$^{-1}$ yr$^{-1}$ [39]. For Monte Carlo simulations, we randomly sampled from normal distributions constructed with the mean and standard deviations noted above.

**Grasslands to cropland.** We used an analysis of grassland conversion to cropland to estimate the loss of grassland to cropland from 2008 to 2012 [43]. Lark et al. [43] estimate grassland loss in Oregon to be 931 ha yr$^{-1}$, using USDA Cropland Data Layer with additional processing and bias correction.

We assume that all perennial root biomass is lost when grasslands are converted to croplands. We used estimates of belowground root biomass from Oregon meadows which found an average of 18.44 Mg C ha$^{-1}$, or 67.6 ± 7.66 MTCO$_2$e ha$^{-1}$ [44]. This estimate is lower than the average used for national-scale analysis of NCS [8]. For Monte Carlo simulations, we sampled from a normal distribution.

## Land management NCS

**Timber harvest.** Following the methodology of Fargione et al. [8], we modeled delayed harvest through the deferment of a percentage of annual harvest in Oregon. In concept, areas of deferred harvest, whether applied on the basis of large even-aged units or to smaller patches, are allowed to grow past the current rotation age and closer to their "biological optimum" (e.g., culmination of mean annual increment) [45,46]. To limit interactions with the risk of wildfire to elevated forest carbon stocks, we did not apply the delayed harvest NCS pathway to forests in counties where more than 50% of forestland was considered at high risk of wildfire [47] (S1 Fig).

We used timber harvest data for each forest ownership class (i.e., private industrial, private non-industrial, state, federal) in Oregon from 2000 to 2016 to estimate baseline harvest rates (available at https://data.oregon.gov/Natural-Resources/Timber-Harvest-Data-1962-2017, S2 Table). We calculated emissions separately for timber harvested east of the Cascades and west

of the Cascades to account for known productivity differences [48]. Harvest emissions were defined as all carbon emitted in the first 20 years following harvest (e.g., committed emissions, *sensu* [8]). Emissions occurring from mill residues used as commercial fuels or 'not used' as well as from wood products that are retired within the first 20 years (i.e., not remaining in use or in landfills after 20 years) were estimated as a percentage of harvest volume and were assumed 'committed emissions' at the time of harvest. Transformed wood products stored beyond 20 years were not included as harvest emissions. As we assume that harvests are deferred for at least 30 years, the biomass included in the deferred harvest is not subsequently harvested within the time period of our study. The climate benefit of deferred harvest is realized during this period with reduced annual harvest. Eventually annual harvest levels are assumed to return to business as usual (BAU), albeit with larger forest carbon stocks [8].

We used published conversion factors to convert harvest volumes from thousand board feet (MBF) to cubic meters [49] and estimated logging residue volume using a residue:roundwood volume ratio of 0.25 calculated from the U.S. Forest Service RPA Assessment [50,51]. Belowground biomass was estimated using the root:shoot ratio of 0.2 [52,53]. We estimated committed emissions from logging residue and belowground biomass to be 56%, following Fargione et al. [8]. Based on 2012 and 2017 USFS RPA Assessments, 15% of harvested biomass becomes unused mill residues or commercial fuel [50,51]. A further 28% of harvested timber volume becomes transformed wood products that are retired and oxidized in the first 20 years [32]. Volumes ($m^3$) were converted to Mg $CO_2$e using specific gravity factors for interior (0.397 g $cm^{-1}$) and coastal (0.423 g $cm^{-1}$) forests (PNW weighted averages for 96% softwood and 4% hardwood harvests; [32]), a carbon fraction of 0.5, and a conversion factor of 44/12 for $CO_2$e to carbon.

For privately-owned, even-aged managed forests (e.g., clearcut harvest) which have a clear difference in annual sequestration rates between recently harvested and a delayed harvest stand [27], we also calculated gains in carbon sequestration associated with delayed harvest scenarios. We estimated the annual area of clearcut harvest on private industrial timberland using global forest change data from 2000–2016 [54] intersected with private industrial timber ownership data. Fire perimeters from Monitoring Trends in Burn Severity (MTBS; mtbs.gov) were used to filter out forest cover loss from wildfire [55]. We estimated the difference in carbon sequestration (MT $CO_2$e $ha^{-1}$) for even-aged managed forests using growth tables for PNW interior and coastal forests after clearcuts [32]. Our estimate of carbon sequestration includes changes in live tree biomass per year from stand ages 0–75, assuming that BAU harvest occurs around stand ages of 45 years [56,57] and harvest extensions increase stand age by at least 30 years before harvest. Therefore, we can calculate the change in sequestration rate ($\Delta C_{seq}$) for extended forest rotations as:

$$\Delta Cseq = Forest\ Growth_{(stands\ 45-75\ years\ old)} - Forest\ Growth_{(stands\ 0-30\ years\ old)}$$

We calculated the $\Delta C_{seq}$ for extending rotations in interior and coastal forests by across all stand types in these productivity regions [32]. The $\Delta C_{seq}$ for interior forests was estimated 30.04 ± 18.03 MT $CO_2$e $ha^{-1}$ while $\Delta C_{seq}$ for coastal forests was estimated as 108.16 ± 21.17 MT $CO_2$e $ha^{-1}$.

**Cover crops.** We used data from the 2012 and 2017 Census of Agriculture to estimate current areal extent of cover crop use (USDA-NASS; available at https://www.nass.usda.gov/Quick_Stats/index.php). The 2012 Census was the first to include a question about cover crops and reported cover crop use increased by 11,170 ha (29%) over the 5-year period, with 48,740 ha planted in cover crops in 2017. Despite the increase, baseline cover crop use represents only

2% of cropland acres in Oregon. We set baseline cover crop use to the 2017 levels and estimated historical variation at 6% per year.

Long-term studies in the inland PNW suggest that crop rotations which include a fallow period have greater soil C loss than those that include a cover crop or diversified crop rotation [58]. In a study of cover cropping achieved through mixed perennial-annual systems in the inland PNW, soil organic carbon (SOC) was estimated to increase by 2.53 $MTCO_2e$ $ha^{-1}$ $yr^{-1}$; however, the authors note a lack of soil organic carbon/cover crop datasets, suggesting uncertainty in applying their estimate outside a narrow geographic region [59]. Nationally, the addition of cover crops is variably estimated to increase soil organic carbon at rates equivalent to 0.37–3.24 $MTCO_2e$ $ha^{-1}$ $yr^{-1}$ [60]. Globally, cover crops are estimated to increase carbon sequestration rates on average $1.17 \pm 0.29$ $MTCO_2e$ $ha^{-1}$ $yr^{-1}$ [61]. We conservatively modeled estimated SOC change using Monte Carlo simulations from a normal distribution following the global meta-analysis with mean = 1.17 and sd = 0.29 $MTCO_2e$ $ha^{-1}$ $yr^{-1}$.

**No-till agriculture.** We used data from the 2012 and 2017 Census of Agriculture to estimate current areal extent of no-till use (USDA-NASS; available at https://www.nass.usda.gov/Quick_Stats/index.php). The 2012 Census was the first to include a questions differentiating no-till, conservation tillage, and conventional tillage and reported no-till use increased by 115,000 ha (40%) over the 5-year period, with 403,280 ha reported as no-till in 2017. In 2012, no-till comprised 30% of 'tilled' cropland in Oregon, while in 2017, no-till comprised 41% of 'tilled' cropland. We set baseline no-till use to the 2017 levels and estimated historical variation at 8% per year.

To estimate the carbon sequestration potential ($MTCO_2e$ $ha^{-1}$ $yr^{-1}$) of no-till agriculture, we reviewed the available literature on tillage practices and soil organic carbon (SOC) with particular focus on the PNW. The majority of regionally relevant studies focused on the inland PNW, east of the Cascades. No consensus exists on the effects of no-till on SOC in the PNW with at least two studies finding no significant effect of tillage on SOC [62,63]. However, in other studies, SOC was estimated to increase by 0.12 to 0.53 $MTCO_2e$ $ha^{-1}$ $yr^{-1}$ when switching from conventional tillage to no-till [59,64,65] and no-till did not negatively affect wheat yield [66]. PNW estimates are on the low end of average soil C sequestration rates, which tend to be lowest in cold northern and arid western states [60,67,68]. We modeled the estimated SOC change using Monte Carlo simulations drawing randomly from a uniform distribution ranging between 0.12 and 0.53 $MTCO_2e$ $ha^{-1}$ $yr^{-1}$.

**Cropland nutrient management.** State-level fertilizer use rates were calculated following published methods [69,70] using annual sales of commercially produced fertilizer from 1997–2015 [71]. Fertilizer sales data were converted from tons of product sold to kg of N, based on the reported chemical composition of the fertilizer [71]. Where composition was not specified for a product, default percentages based on the product's reported fertilizer code were used. From these data, we calculated current use of N fertilizer at 186,294 Mg N $yr^{-1}$ with a historical variation of 18%.

Ribaudo et al. [72] suggest that best management practices (BMPs) for nitrogen application include limiting nitrogen application to no more than 40% more than that removed by crops at harvest. We estimated the ratio of N removed by crops to N fertilizer used for each county in Oregon using nutrient balance data calculated by the International Plant Nutrition Institute [73]. IPNI publishes county level N ratio estimates from 1987 to 2014; we found that counties comprising 40% of the cropland in Oregon exceed the recommended nutrient use efficiency. Thus, we assume that nitrogen reductions could be applied to 40% of cropland in our scenarios.

Emissions of nitrous oxide ($N_2O$) are strongly correlated with fertilizer N rate [74,75]. The Intergovernmental Panel on Climate Change (IPCC) Tier 1 methods for GHG inventories

assume a total emissions factor (EF) for $N_2O$ to be 1.1 to 1.3% of the N inputs [76]. However, studies suggest that the EF can be even higher at N input levels that exceed crop demand for N [74,77,78] and a recent analysis of historical N-flux from agriculture estimated total EF to be 2.54% [75]. Here, we incorporated uncertainty in the $N:N_2O$ EF using Monte Carlo simulations to draw the total EF from a uniform distribution ranging from 1–2.54%. For each simulation, used the selected total EF to translate N fertilizer use to $N_2O$ emissions. Finally, we multiplied the resulting $N_2O$ emissions by 298 to calculate $CO_2e$.

## Restoration NCS

**Reforestation after wildfires.** Reforestation after wildfires is defined as replanting without salvage harvesting or site preparation after moderate (25% - 75% basal area mortality) to severe (75% - 100% basal area mortality) wildfires. We limited the post-wildfire reforestation pathway to federal lands because the Oregon Forest Practice Act (OFPA) has strong requirements mandating replanting on private lands after planned and post-wildfire harvests [79]. Furthermore, we assumed that private landowners of substantial forestland area would typically conduct salvage harvests post-wildfire and be required to replant forests under the OFPA and/or would have preexisting financial incentives to replant. The US Forest Service (USFS) and Bureau of Land Management (BLM) manage over 7 million ha of federal forestlands in Oregon.

To estimate the current replanting effort on federal land, we first calculated the average annual area available for replanting using wildfire severity data and management objectives on USFS and BLM land (e.g., no active reforestation within wilderness area boundaries). Specifically, we considered areas that burned at moderate or high severity between 2000 and 2015 since these are likely to be replanted. Wildfire severity was based on data from Monitoring Trends in Burn Severity (MTBS; mtbs.gov) and reclassified with consistent, ecologically informed fire severity thresholds and to account for unburned areas [80]. We further limited areas available for replanting to land managed as 'active', 'multiple objective', and 'stand-age dependent' management on USFS land and 'active' and 'multiple objective' management on BLM land [81,82]. We used publicly available datasets to calculate the average areal extent of post-wildfire reforestation on USFS and BLM land [83,84]. Finally, we calculated the annual rate of postfire reforestation as the proportion:

$$P_{replanted} = \frac{mean\ annual\ area\ replanted}{mean\ annual\ burned\ area\ available\ for\ replanting}$$

We assumed that replanted vegetation on federal lands would be similar to the potential vegetation type, which we extracted from LANDFIRE Biophysical Setting (BPS) data [85]. We used USFS yield tables to estimate carbon sequestration rates for each BPS forest type using a crosswalk based on spatial overlap and cover type name similarity [8,32]. We further classified forest types into three broad productivity classes based on expected C storage in the first 35 years using Jenk's natural break classification. For each of the three productivity classes, we produced tables varying C sequestration by stand age based on published growth tables [32]. Natural regeneration after moderate and high severity wildfires in the PNW can be limited by a lack of seed source [86] and regeneration can be delayed due to environmental conditions following wildfire [87]. Forests in the PNW that experience wildfires are likely to naturally regenerate, but with slower initial growth rates and uneven spatial coverage than replanting after wildfire [88–90]. We assume that replanting occurs within the first 2 years post-wildfire, while natural regeneration is delayed at least 10 years. Therefore, we calculate the annual C sequestration rate for each reforestation using the following equation, where $C_{planted}$ is the

carbon sequestration rate for replanted forests and $C_{natural\ regeneration}$ is the expected natural regeneration sequestration for a particular stand age, which is set to zero for years 0–10 following wildfire:

$$Reforestation\ \Delta Sequestration = C_{planted} - C_{natural\ regeneration}$$

For each Monte Carlo simulation, we incorporated uncertainty in wildfire area and reforestation sequestration rates. We assumed that the distribution of burned area of moderate to high severity on federal land for the period of our simulation (2020–2050) would not change from observed (2000–2015). We modeled wildfire area available for replanting using Monte Carlo simulations from a normal distribution based on historical means and standard deviation of fire areas in low, medium, and high productivity forests. To quantify uncertainty of our reforestation sequestration rate, we assumed a normal distribution with the mean equal to the reforestation sequestration as calculated above and uncertainty of ± 20%.

**Tidal wetland restoration.** We defined tidal wetland restoration as restoring tidal processes in areas where tidal wetlands were the historical natural ecosystem. We estimated the current annual rate of tidal wetland restoration using data from the Oregon Watershed Enhancement Board (OWEB) which compiles data on restoration project objectives and areas including estuary restoration projects from 2000 to 2017 [91]. Between 2000 and 2017, 880 ha of estuarine tidal wetlands have been restored resulting in a baseline implementation rate of 48.9 ha yr⁻¹. In addition, we estimated the total area available for tidal wetland restoration by combining data on tidal-influenced wetlands, tidal impairment, and historical tidal wetland extant [92,93]. We limited tidal restoration opportunity to the areas with the highest salinity to exclude freshwater and mesohaline wetlands with high rates of methane release [94–96]. The resulting area included 5205 ha of tidal wetland restoration opportunity, which served as the upper threshold for cumulative restoration area.

We modeled carbon benefit attributable to tidal wetland restoration by estimating the increase in sequestration as well as the avoided GHG emissions from drained and degraded marshes. Restored tidal wetlands carbon sequestration varies from 0.79 to 0.94 MT C ha⁻¹ yr⁻¹ on the west coast and PNW [97–99]. We estimated carbon sequestration to be the average of reported values, 0.87 MT C ha⁻¹ yr⁻¹, or 3.17 ± 0.39 MTCO₂e ha⁻¹ yr⁻¹. Altered water salinity and water table elevation can influence the emissions of methane ($CH_4$) [100,101]. We estimated the avoided loss of $CH_4$ to be 0.23 Mg $CH_4$ ha⁻¹ yr⁻¹, or 5.66 ± 3.53 MTCO₂e ha⁻¹ yr⁻¹ [101]. For Monte Carlo simulations, we drew samples from a normal distribution with a mean and standard deviation of 8.84 ± 3.92 MTCO₂e ha⁻¹ yr⁻¹ to characterize the carbon benefits due to tidal wetland restoration.

**Riparian reforestation.** We define riparian forest restoration as conversion from non-forest to forest along riparian areas. We estimated the annual rates of riparian forest restoration using data on reported riparian restoration tree plantings from 2001 to 2017 [91,102]. Data included voluntarily reported data from projects that included funding from the Oregon Watershed Enhancement Board (OWEB), which provides matching funds for riparian restoration projects across a variety of ownerships statewide, as well as reported data from the Oregon Conservation Reserve Enhancement Program (CREP), which provides funds for eligible conservation practices on agricultural lands. Because riparian tree plantings may occur outside of those funded by OWEB and CREP, we consider our annual area estimate to be conservative (i.e., likely underestimates the overall annual restoration area). We estimated the baseline annual riparian reforestation to be 1713 ha yr⁻¹ in interior Oregon and 683 ha yr⁻¹ in coastal/western Oregon.

We estimated the maximum extent of riparian reforestation opportunity by combining published floodplain maps [103] with recent mapped tree canopy cover data [104] and environmental site potential [85]. We considered areas to have riparian reforestation potential if they (1) occurred within a 100-year floodplain, (2) had less than 40% canopy cover, and (3) had ecological site potential that included forest, woodland, or was undetermined based on biophysical setting. Urban areas are outside the scope of this study and so we did not include areas mapped as high or moderate density development [34] in our estimate of riparian reforestation potential. The resulting area included 202,415 ha of riparian reforestation opportunity, 70% of which is located in coastal/western Oregon. These estimates served as the upper threshold for cumulative restoration area in our implementation scenarios.

Carbon accounting methods for restored (i.e., planted) riparian forests and woodlands have not been well-developed in the literature [105–107]. Thus, we estimated the difference in carbon sequestration for restored riparian forests using growth tables for afforestation in PNW interior and coastal forests [8,32]. In Oregon riparian restoration projects, the species used in tree plantings varies by geographic region [91]. The majority of planted conifers in western Oregon is comprised of Douglas-fir (*Pseudotsuga menziesii*) mixed with western red cedar (*Thuja plicata*), western hemlock (*Tsuga heterophylla*), and Sitka spruce (*Picea sitchensis*) while eastern Oregon riparian conifer plantings include Ponderosa pine (*Pinus ponderosa*). Hardwood riparian plantings include willow (*Salix* spp.), alder (*Alnus spp.*), maple (*Acer spp.*), and cottonwoods (*Populus spp.*). We estimated sequestration rates separately for conifer only, hardwood only, and mixed plantings. For each of the riparian planting types, we calculated mean annual sequestration from stand ages 0 to 30 years, which is a time span relevant to climate mitigation needs and matches our simulation length. We calculated weighted averages for interior and coastal plantings using the proportion of hardwood, conifer, and mixed plantings reported in each region. We added an additional soil carbon accumulation rate of 0.09 MT C ha$^{-1}$ yr$^{-1}$, following Fargione et al. [8] and based on published soil carbon accumulation rates in reforestation [108]. We estimate that riparian reforestation sequesters 12.17 ± 0.43 MTCO$_2$e ha$^{-1}$ yr$^{-1}$ and 15.81 ± 0.45 MTCO$_2$e ha$^{-1}$ yr$^{-1}$ in interior and coastal riparian plantings, respectively. For Monte Carlo simulations, we sampled from normal distributions constructed with these estimates for the mean and standard deviation of riparian reforestation sequestration.

**Sagebrush-steppe restoration.** To estimate the recent implementation rate of restoration projects, we queried the Conservation Efforts Database for completed sagebrush-steppe restoration projects in Oregon for the years 2009–2014 [109]. To further refine the estimate, we also queried the Land Treatments Digital Library for 'seeding' treatments by the BLM, which manages the majority of sagebrush-steppe habitat in Oregon [110]. Combined, these queries resulted in an estimate of attempted sagebrush-steppe restoration actions on 55,900 ha yr$^{-1}$. Restoration of sagebrush-stepped ecosystems has proven to be very difficult and success varies based on the elevation and moisture gradients as well as dominance of invasive species [111,112]. There is no published rate of restoration success across SE Oregon, so we relied upon expert opinion to estimate current restoration success rates as 10% (J. Kerby, *personal communication*), thus setting our baseline to 5,590 ha yr$^{-1}$. To set an upper limit on restoration activities for our scenarios, we estimated the total area of invasive annual grass dominated sagebrush-steppe ecosystems at 906,000 ha, using published datasets [38].

We estimated that restored sagebrush-steppe would increase carbon sequestration as compared to invasive annual grass at 0.81 ± 0.44 MTCO$_2$e ha$^{-1}$ yr$^{-1}$, the same rate as foregone carbon sequestration due to conversion from sagebrush-steppe to annual grasses [39]. For Monte Carlo simulations, we sampled from a normal distribution with the mean and standard deviation specified above.

## NCS implementation scenarios and uncertainty

We evaluated the potential for NCS to provide carbon benefits under multiple implementation scenarios. We conducted three scenarios where we varied the implementation rate for each NCS pathway (Table 3) and calculated the annual and cumulative GHG reductions possible from NCS implementation as compared to the current baseline implementation. All three scenarios included a ramp up period rather than assuming that implementation of NCS increased immediately to target levels in 2020. The *Limited Implementation* scenario allowed NCS implementation to ramp up for a ten-year period from 2020 to 2030 and then remain stable after 2030. Each NCS activity was implemented at a rate equal to the relative level of variation (i.e., coefficient of variation) in its implementation over the past 10 to 20 years. Where historical implementation rates were not available, we assumed a 10% change from baseline implementation. The *Moderate Implementation* scenario constrains the implementation of NCS activities to feasible yet aggressive levels based on stakeholder feedback. In this scenario, we allowed NCS implementation to ramp up over a period of 10 to 30 years. The *Ambitious*

**Table 3. Implementation scenarios for natural climate solution activities.** All scenarios are expressed as percent change from the baseline rate of each activity.

| | Natural Climate Solution Activity | Scenario Implementation Rates (% change from baseline) | | |
| --- | --- | --- | --- | --- |
| | | Low Implementation | Moderate Implementation | Ambitious Implementation |
| **Avoided Conversion** | Avoided conversion of forests to rural development | Reduced conversion by 10%, allows 10 years to reach target | Reduced conversion by 50% by 2030, keeps 50% of baseline conversion rate 2030–2050 | Reduced conversion by 100% (i.e., zero hectares converted) by 2030 and maintains zero conversion rate 2030–2050 |
| | Avoided conversion of forests to urban development | Reduced conversion by 10%, allows 10 years to reach target | Reduced conversion by 50% by 2030, keeps 50% of baseline conversion rate 2030–2050 | Reduced conversion by 100% (i.e., zero hectares converted) by 2030 and maintains zero conversion rate 2030–2050 |
| | Avoided conversion of sagebrush-steppe to invasive annual grasses | Reduced conversion by 10%, allows 10 years to reach target | Reduced conversion by 10% by 2030, and further reduced to 20% of baseline in 2030–2050 | Reduced conversion by 30% (i.e., zero hectares converted) by 2030 and that conversion rate 2030–2050 |
| | Avoided conversion of grasslands to tilled cropland | Reduced conversion by 10%, allows 10 years to reach target | Reduced conversion by 50% by 2030, and further reduced to 100% of baseline (i.e., zero hectares converted) by 2050 | Reduced conversion by 100% (i.e., zero hectares converted) by 2030 and maintains zero conversion rate 2030–2050 |
| **Land Management** | Deferred timber harvest | Reduced timber harvest equivalent to the historical variation in timber harvest over the last 20 years on each forest ownership; allows 10 years to reach reduction levels. Reductions are 10–100% of baseline depending on ownership (see S2 Table). | Gradual reduction of timber harvest to target reduction of 75% on most ownerships and 15% on State and Private Industrial ownerships by 2030. Allows 73% of current harvest volume (overall) from 2030–2050 | Gradual reduction of timber harvest to target reduction of 100% on most ownerships and ~20% on State and Private Industrial ownerships by 2030. Retains 60% of current harvest volume (overall) from 2030–2050 |
| | Use of cover crops | Increased cover crop use by 40% over the next 10 years and then steady after 2030 | Increased cover crop use by 150% in 2030. Continued increase to quadruple cover crop use by 2050 | Continual increase in cover crop use to reach 50% of all cropland in 2050 (equal to a 25x increase in cover crop use) |
| | Natural Climate Solution Activity | Scenario Implementation Rates (% change from baseline) | | |
| | | Low Implementation | Moderate Implementation | Ambitious Implementation |
| **Land Management** | No-till agriculture | Increased no-till use by 30% over the next 10 years and then steady after 2030 | Increased no-till use by 100% in 2030. Continued increase to 150% of baseline by 2050 | Continual increase in no-till use to reach 100% of 'tilled' cropland in 2050 (equal to 3x increase in no-till use) |
| | Nutrient management | Reduced N-fertilizer use by 20% on half of fertilized acres by 2030, steady after 2030 | Reduced N-fertilizer use by 25%, implementing on half of fertilized acres by 2030 and the remaining half by 2050 (25% total reduction by 2050) | Reduced N-fertilizer use by 40% on all acres by 2030 and maintained this decrease through 2050 |

*(Continued)*

**Table 3.** (Continued)

| Restoration | Replanting after wildfire on federal land | Increased replanting rate by 100% over the first 10 years, steady after 2030 | Gradually increased replanting rate by 100% in 2030 and 150% baseline by 2050 | Rapidly increased replanting rate by 100% in 2030, 300% in 2040, and 700% of baseline in 2050. In 2050, 63 to 84% of wildfire area replanted each year. |
| --- | --- | --- | --- | --- |
| | Riparian forest restoration | Increased riparian reforestation by 72% in eastern OR and 42% in western OR (based on historical variation); allows 10 years to reach target rates | Doubling (100% increase) in riparian reforestation rate by 2050, 250% increase by 2040, and 300% increase by 2050. | Rapidly increased riparian reforestation rates to reach the maximum area available by 2050. Maximum area estimated as 76,635 ha east of the Cascades and 125,780 ha west of the Cascades. East side riparian reforestation increases to 500% by 2030 and reaches the maximum area threshold in 2032. West side riparian reforestation increases by 1000% (10x) by 2030 and reaches maximum area threshold by 2044. |
| | Tidal wetland restoration | Increased tidal wetland restoration by 100% in 2030 and maintain this increase through 2050 | Increased tidal wetland restoration by 200% in 2030 and maintain this increase through 2050 | Rapidly increased tidal wetland restoration by 100% in 2030, 300% baseline in 2040, and 700% in 2050; reaches maximum area of cumulative restoration of 5200 ha by 2048. |
| | Invasive annual grasses to sagebrush-steppe | Increased sagebrush-steppe restoration by 10% in 2030 and maintain this increase through 2050 | Increased sagebrush-steppe restoration by 100% in 2030 and by 200% by 2050 | Increased sagebrush-steppe restoration by 200% in 2030 to 11,180 ha $yr^{-1}$ and maintained this increase through 2050. Cumulative restoration is 251,000 ha. |

*Implementation* scenario assumes aggressive implementation of NCS which allows NCS implementation to ramp up quickly for at least a ten-year period. Avoided conversion pathways reached zero conversion after 10 years whereas restoration pathways increased rapidly toward the maximum area available for restoration. In all scenarios, we assumed that the current extent of cropland and grazing areas is maintained and that the maximum annual timber harvest deferment does not exceed 25% of baseline harvest levels on private industrial forestland and does not decline below 40% of current overall harvest levels.

# Results

## Annual reductions by 2035 and 2050

We found that, under Ambitious Implementation, combined NCS pathways could achieve median greenhouse gas (GHG) reductions up to $8.3 \pm 1.3$ MMTCO$_2$e $yr^{-1}$ by 2035 and further contribute to reductions up to $9.8 \pm 1.7$ MMTCO$_2$e per year by 2050 (Fig 2). Under the Limited and Moderate scenarios, combined NCS pathways were estimated to provide reductions of $2.7 \pm 0.3$ and $4.5 \pm 0.7$ MMTCO$_2$e $yr^{-1}$ by 2035 and further reductions of $2.9 \pm 0.4$ and $5.5 \pm 0.9$ MMTCO$_2$e $yr^{-1}$ by 2050. Estimated reductions from each NCS activity ranged from 0.003 to 5.2 MMTCO$_2$e $yr^{-1}$, depending on the specific activity and scenario implementation (Fig 3, S3 Table). Land management pathways achieved the greatest reductions in GHG (70 to 90% of overall reductions), with deferred timber harvest comprising the largest reduction. The relative contribution of other NCS activities increased with increasing implementation rates in the Moderate and Ambitious Scenarios. For example, cover crops comprise less than 1% of CO$_2$e reductions in the Limited scenario but contributed 11% of the overall CO$_2$e reductions under Ambitious implementation. Avoided conversion pathways provided 2 to 9% of potential reductions, while restoration pathways comprised 6 to 21% of annual reductions depending on the implementation scenario.

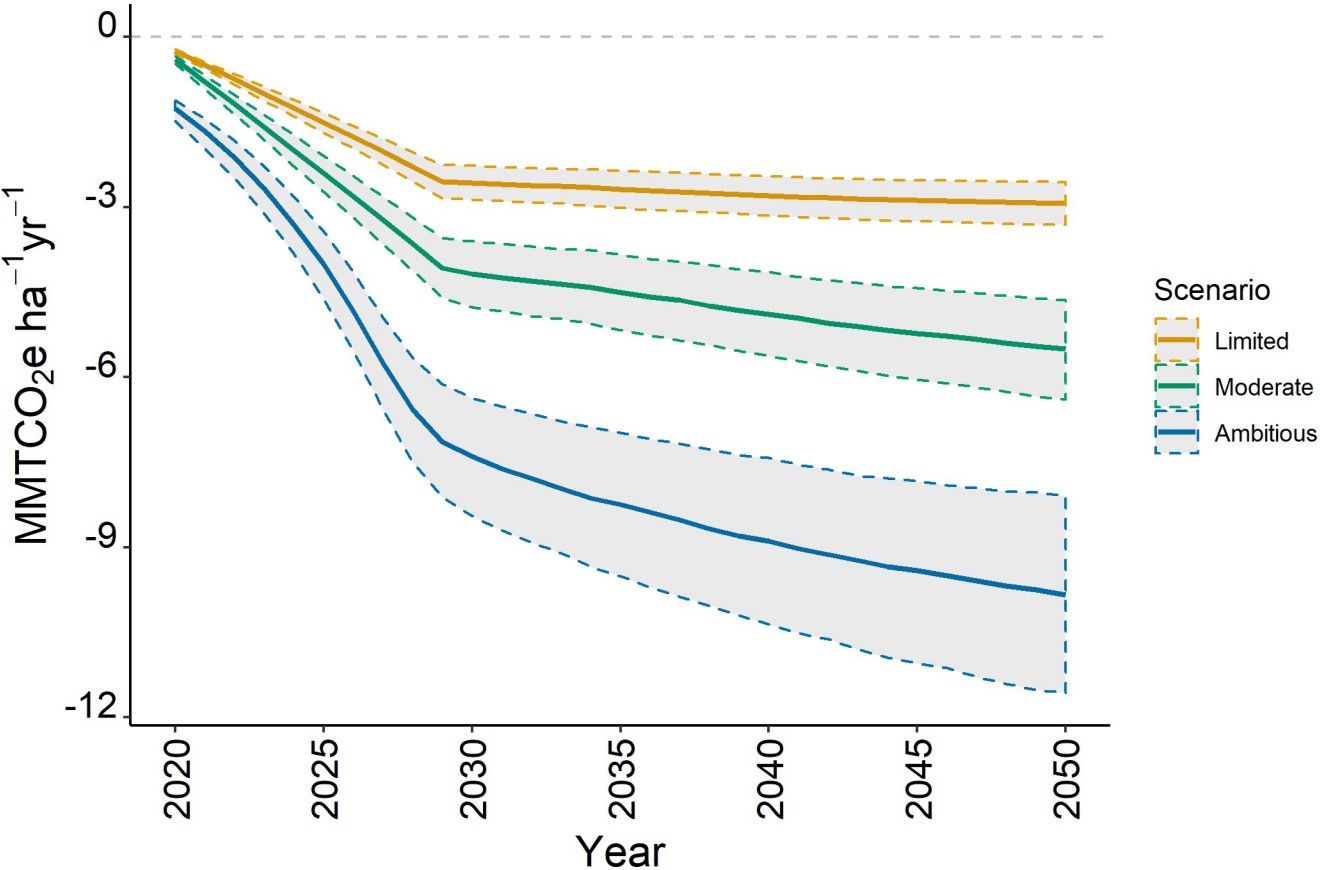

**Fig 2. Annual greenhouse gas reductions in million metric tons of CO₂e from all 12 NCS activities combined under three implementation scenarios from 2020 to 2050.** Dotted lines show the 90% confidence interval around the median estimated reduction for each scenario.

## Cumulative reduction by 2050

Cumulative GHG reductions over the 30-year period of our analysis, reported as the median with 90% confidence interval lower and upper estimates in parenthesis, ranged from 72.2 (63–81) to 222 (184–260) MMTCO₂e in the Limited and Ambitious scenarios, respectively (Fig 4). The Moderate scenario resulted in cumulative reductions of 123 (104–142) MMTCO₂e by 2050. As with the annual reductions, deferred timber harvest comprised the largest cumulative reduction by 2050 in all scenarios (85%, 73%, and 61% of the Limited, Moderate, and Ambitious scenarios, respectively). Other forest pathways, i.e., avoided conversion, riparian reforestation, and replanting after wildfire, comprised 10 to 17% of cumulative reductions while agricultural pathways, i.e., cover crops, nutrient management, and no-till, comprised 3% to 12% of cumulative reductions. Sagebrush-steppe, grassland, and tidal wetland pathways contributed the least to cumulative reductions (1 to 4%).

The relative share of the GHG reductions due to timber deferment varied across property ownership, and absolute reductions depended on the baseline annual timber harvest for each ownership (S2 Table). Despite comprising 72% of the baseline timber harvest, private industrial ownership comprised between 40 to 49% of avoided emissions from timber deferment (S2 Fig). Private non-industrial ownerships, which accounted for 9.3% of baseline harvest and were assumed to have zero harvest after 2030 in the most ambitious scenario, comprised 20 to 26% of the cumulative GHG reduction from avoided emissions due to timber deferment.

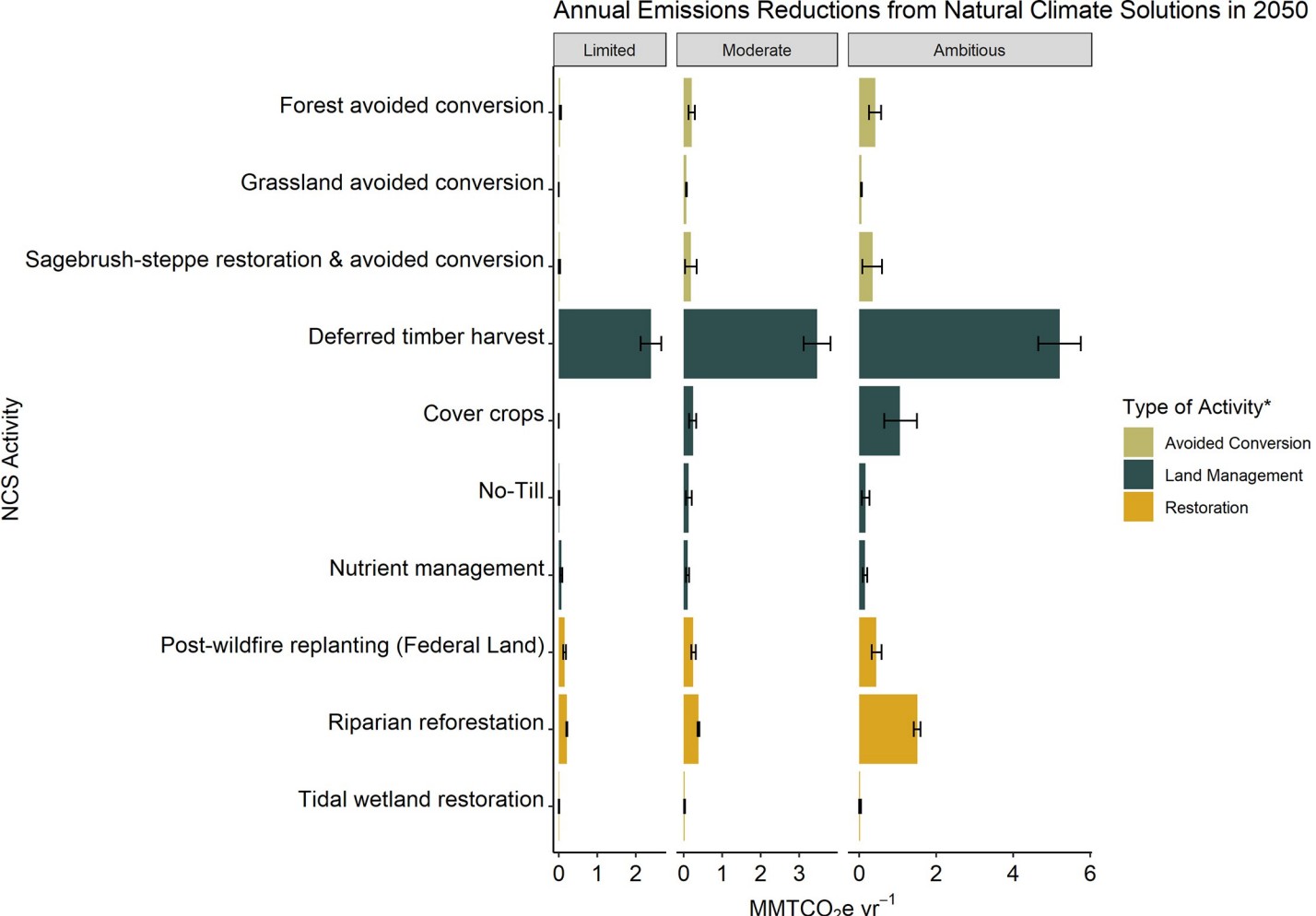

**Fig 3. Estimated annual reductions in MMT CO₂e for each NCS activity under three different implementation scenarios in year 2050.** Error bars represent the 90% confidence interval around the median value from simulations. Activities are grouped: avoided conversion (beige), land management (dark green), and restoration (orange).

Similarly, Federal ownerships, which comprise 9.2% of baseline harvest rates, provided 23 to 26% of the GHG reductions from timber deferment. State, local, and tribal ownerships comprised the remainder of the cumulative GHG reductions associated with timber deferment.

## Contribution to Oregon's GHG emissions targets

In 2017, Oregon's statewide emissions, which are calculated by accounting for emissions from agriculture, industrial, residential/commercial, and transportation sectors, were estimated at 64 MMT CO₂e [113]. In order to meet the State's GHG reduction targets for 2035 and 2050, GHG emissions need to be reduced by 30.1 and 50 MMT CO₂e in the next 15 to 30 years. The combined NCS activities could provide 9%, 15%, and 27% of the needed reductions in year 2035 and 6%, 11%, and 19% in 2050 under Low, Moderate, and Ambitious Scenarios, respectively (S2 Table; Overall annual reductions). If emissions reductions and fossil fuel mitigation in other sectors are used to reach the State's 2050 emissions target of 14 MMT CO2e, NCS could contribute 21 to 69% of the additional annual GHG reductions needed to reach zero emissions (Fig 5).

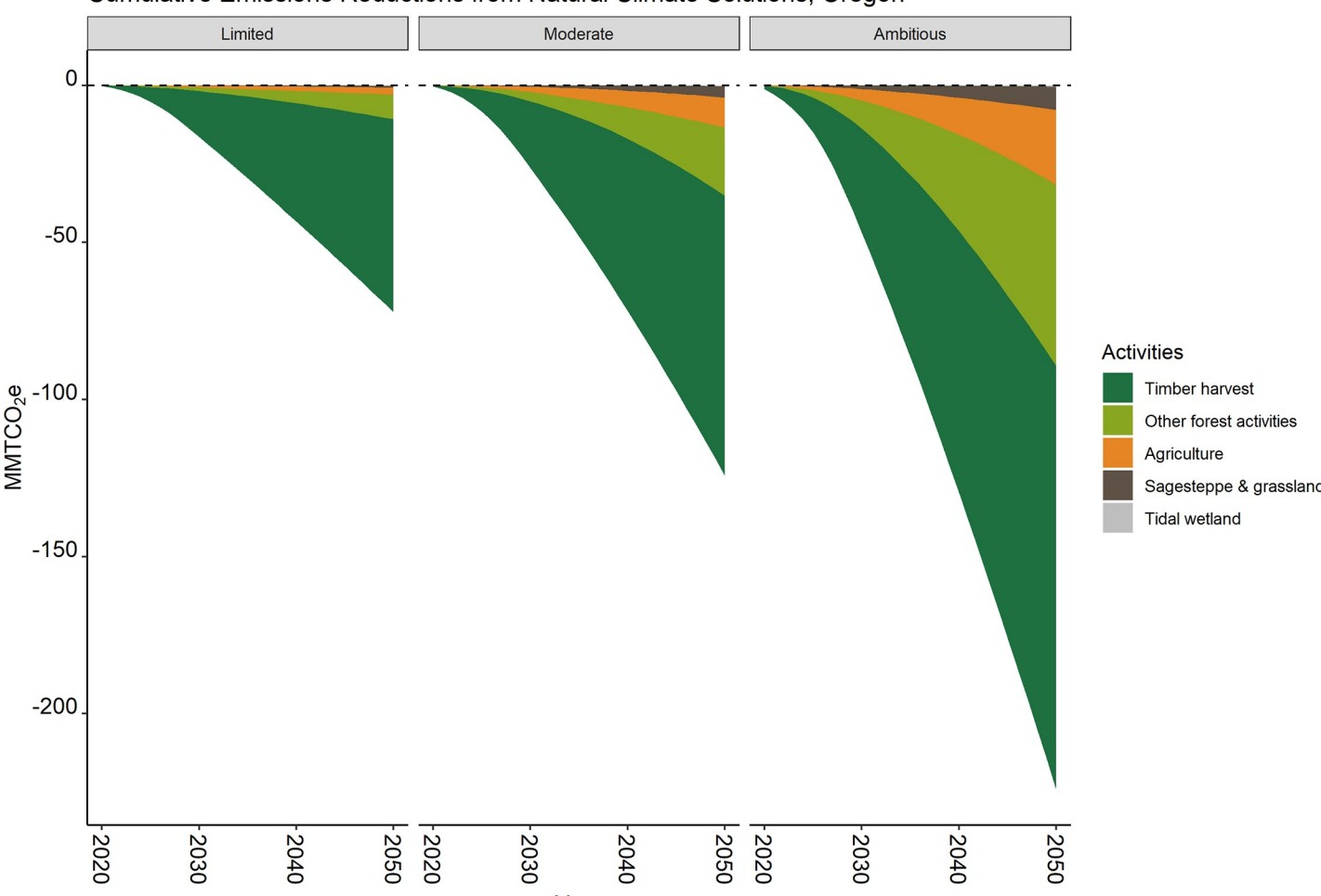

**Fig 4. Cumulative GHG emission reductions from NCS activities in Oregon under three implementation scenarios.** Results illustrate the large contribution from deferred timber harvest (dark green) as compared to other forest-based activities (light green), agricultural activities (orange), sagebrush-steppe and grassland (brown), and tidal wetlands (grey).

## Discussion

To limit the most serious of impacts from climate change, society needs to act quickly to reduce GHG emissions and drawdown GHGs in the atmosphere [1]. Subnational commitments to limit GHGs are increasingly common, including in Oregon and other states participating in the U.S. Climate Alliance. In this study, we found that Oregon could achieve additional GHG reduction through NCS activities such as changing land management practices, restoring native ecosystems, and avoiding conversion of native habitats. Specifically, we found that increased implementation of NCS activities could reduce GHG emissions by 2.9 to 9.8 MMT $CO_2$e yr$^{-1}$ and contribute 6 to 20% of the GHG emissions mitigation needed to reach Oregon's current emissions goal of 14 MMT $CO_2$e by 2050.

Rising scientific consensus indicates that net emissions of $CO_2$ must fall to zero for temperatures to stabilize and to avoid the most catastrophic climate change impacts [1,114–116]. Our results suggest that increased investments in land management, restoration of ecosystems, and avoided conversion of native habitats can enhance the land sector's ability to act as a carbon sink and achieve GHG reductions beyond fossil fuel mitigation alone. Assuming aggressive

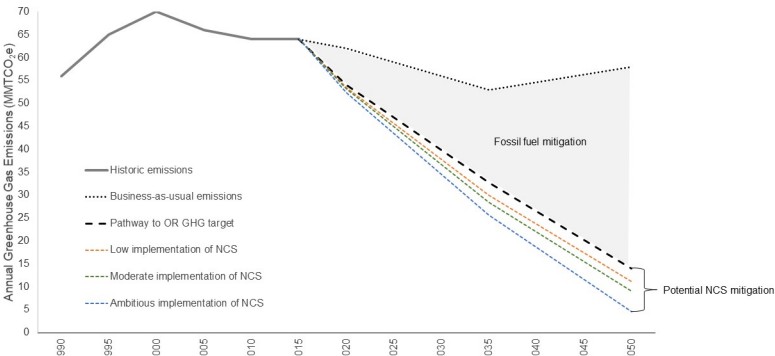

**Fig 5. Contributions of NCS activities to Oregon's GHG reduction goals.** Shown are the historic emissions before 2017 (solid gray line), the projected business-as-usual emissions trajectory (dotted black line, OR Global Warming Commission 2018), and the pathway to reach Oregon's GHG reduction goals for 2035 and 2050 (dashed black line). The grey area shows the needed fossil fuel mitigation across other sectors while the colored dashed lines show the potential contribution of NCS under low (orange), moderate (green), and ambitious (blue) implementation.

fossil fuel mitigation and emissions reductions from other sectors can meet the target of 14 MMT $CO_2$e by 2050, we found that NCS activities have the potential to provide 20% to 70% of the additional GHG reduction needed to reach zero emissions in 2050.

The scenarios we explored in this study are consistent with increasing investments or otherwise increasing the implementation of NCS at limited, moderate, and ambitious levels above current practice. In all scenarios, improved land management strategies provided the greatest combined potential GHG reductions, followed by restoration activities. In contrast to other studies [7,8,20], avoided conversion activities contributed least (between 2 and 10%) to the potential annual GHG reduction benefit for the state of Oregon. Oregon's statewide land-use planning program, instituted in 1973, limits development to areas within urban growth boundaries resulting in lower conversion rates of forest and agricultural lands to urban and suburban as compared to regional and national trends [25]. Avoided conversion of forests to other land uses has been cited as one of the most important NCS pathways globally [7], one of the lower-cost NCS opportunities nationally [8], and contributed 10–15% of the assessed annual mitigation potential in California [20]. The pre-existing limits to conversion of natural and working lands in Oregon provide estimated GHG benefits of 1.7 MMTCO$_2$e per year [117] and create important differences in the potential for additional GHG reduction benefits from avoided conversion NCS at the state level, where the carbon storage benefits of avoided conversion have already been realized.

We found that deferring timber harvest, i.e., delaying a portion of annual timber harvest each year, has the single largest mitigation potential for any NCS activity in the state of Oregon (2.3–5.2 MMT $CO_2$e yr$^{-1}$). Forests cover a large area of Oregon and trees store large amounts of carbon per unit area. Oregon's forests, particularly in the West Cascades and Coast Range ecoregions, are some of the most naturally carbon-rich forests in the world but currently store carbon volumes much less than their ecological potential [44]. In the PNW, older forests store significantly more carbon than younger forests [118]. Moreover, much of the carbon removed from forests during harvest is lost to the atmosphere shortly after harvesting [32], thus deferring timber harvest results in substantial carbon benefits both by keeping stored carbon in the forest and by allowing continued sequestration, which can be relatively low in the initial years following clearcut or regeneration harvest [119]. Deferred timber harvest can be achieved through multiple mechanisms ranging from lengthening harvest cycles or changing harvest strategies to partial harvest and alternative management on forestlands [120,121]. In addition

to reducing the near-term carbon emissions, managing for longer rotations and more diverse forest structure would result in long-term increases to in-forest carbon stocks [121–128].

Our finding that timber harvest management provides that greatest potential greenhouse gas reduction is consistent with recent published assessments focused on Oregon's forests [127,129]. Law and colleagues [127] simulated the effect of protecting existing forests, lengthening harvest cycles, re- forestation, afforestation, and bioenergy production with product substitution on net ecosystem carbon balance (NECB) across the state of Oregon and found that lengthening harvest cycles on private land and restricting harvest on public lands resulted in the greatest increases in NECB. Importantly, forest management targeted at preserving high carbon stores can also result in protection of biodiversity [130]. Our study provides additional evidence that forest management in Oregon's productive forests can lead to meaningful state-level GHG emission reductions.

While we specifically model timber harvest deferral, altering other aspects of the timber harvest and wood processing system could also result in emission reductions [131]. In our study, we assume that 15% of the annual harvested wood volume results in unused mill residue or mill residue burned on site [50,51]. The remainder is assumed to be the transformed wood products pool, of which 72% is allocated to long-term storage (i.e., the carbon remains stored in these products for 20 years or more) [*sensu* 8,51]. Increasing the proportion of transformed wood products in the long-term storage pool, for instance through increasing the allocation of current harvest to durable timber products like mass-timber building materials, may provide one viable option for reducing overall harvest emissions. However, shifting wood product pools is unlikely to result in GHG emission reductions at the same order of magnitude as increasing rotation lengths and managing for older, more diverse forests [46,122,123]. Product substitution, which assumes the use of wood products materials in place of more emission intensive alternatives, has been treated variably in carbon accounting assessments [131]. While product substitution may provide GHG emission reductions, it is not included in our study due to the large and compounding uncertainty in assumptions related to estimating substitution [46,123].

In the counties we considered eligible for timber harvest deferment (i.e., less than 50% of forest cover at high risk of wildfire), private industrial forest landowners supply over 70% of the baseline harvested timber volume. Private industrial forestland owners include forest product companies, Timber Investment Management Organizations (TIMOs) and Real Estate Investment Trusts (REITs). State, federal, and non-industrial private forest landowners provide a further 30% of the harvest volume. Our scenarios limit overall timber harvest reductions, particularly on private industrial forests, to maintain harvests at no less than 80% of current levels in the Limited Scenario and 60% of current levels in the Ambitious Scenario (S2 Fig). In our study, modeled changes to private industrial timber harvest provided 40 to 46% of GHG emissions reductions from harvest deferment while changes in non-industrial private forest timber harvest comprised over 20% of the GHG emissions reductions. Opportunity costs of restricting harvest can be substantial for some forest landowners [121,132–134] and in particular, TIMOS and REITS, which tend to prioritize revenue generation, may have less flexibility than small non-industrial private forest landowners to change management [134,135]. More research is needed to determine the incentive, policy, and market conditions under which different landowners are able and willing to participate in timber deferment programs.

Despite its high potential, timber harvest deferment may be a challenging NCS to implement given current socio-economic realities within the state [136–140]. Analyses have suggested that timber deferment equivalent to these levels may be possible under increased carbon pricing (e.g., $50 to $60 per tCO2e; [132,141,142]) but the price of carbon on the

voluntary and compliance markets is well below economic returns available from harvest [132]. In addition, Oregon is one of the largest suppliers of softwood timber in the United States and the timber and forest related sector comprises an important part of the economy in Oregon, particularly in rural communities. To be included as a successful NCS strategy, the development of equitable and acceptable incentives for deferred harvest on private forest lands will need to consider and mitigate potential impacts to rural communities and tradeoffs for forest sector stakeholders [143–146].

Riparian reforestation provides the second largest mitigation potential by 2050 under moderate and ambitious implementation and has the highest carbon sequestration per unit area. The carbon sequestration estimates used in this study for riparian reforestation are lower than the sequestration rates reported in literature on natural regeneration of riparian areas in parts of the Pacific Northwest [147,148] but we lack published data from restored riparian forests across a range of conditions in the PNW [106,107]. In a recent review, Dybala and colleagues [107] found that planted riparian forests had faster initial rates of C sequestration than naturally regenerating counterparts. Thus, our study may underestimate the GHG benefit of riparian reforestation. Riparian reforestation is often targeted with restoration goals aimed improved fish habitat, floodplain connectivity, and water quality [106,149,150] and has wide-ranging support through established incentive and granting structures [151–153]. The existing programmatic structure in Oregon, along with the substantial co-benefits associated with riparian reforestation and areal extent of the opportunity [154], suggest that realizing the carbon benefits from this NCS activity may be relatively easier than timber harvest deferment.

Under Ambitious implementation, changes in agricultural management could reduce 1.39 MMT $CO_2$e of GHG emissions annually by 2050. These GHG emission reductions are primarily attributed to increased cover crops. In many cases, cover crops bring additional benefits including controlling nitrate leaching, providing nutrients especially through nitrogen fixation, conserving water, and improving soil quality [61,155–158]. Despite evidence that cover crops can provide both environmental and yield benefits, less than 2% of Oregon's total cropland is planted to cover crops under baseline conditions. Cover crops may provide an achievable route to increasing carbon storage with ample opportunity for increased adoption through cost-share assistance and highlighting successful local examples of cover crop use [159]. Cover crops are typically grown in combination with main summer annuals (e.g., corn and spring cereals) as a winter rotation or can be used to eliminate summer fallow in fall and winter crops such as winter and spring wheat [157]. Cover crops can be used in Oregon as either an additional crop to replace fallow periods between main crops or as inter-row cover in specialty crops such as orchards, berries, and hops. In our most ambitious scenario, we increased use of cover crops to 50% of cropland. With increased incentives and payment for ecosystem services, cover crops may be possible over even larger areas and provide increased GHG reductions.

For tidal wetland restoration in Oregon, relative contribution to GHG reductions is limited by the applicable geographic extent. Conversion of tidal marshes to pastureland or agriculture, primarily through construction of dikes, is the primary human-caused change to Oregon's tidal wetlands [160]. However, since the 1970s, state and federal policies have limited further conversion of tidal marshes resulting in a negligible annual conversion rate [161]. Despite having high carbon sequestration potential per unit area [162], the overall GHG reduction potential is relatively small for tidal wetlands because increased restoration will saturate the available area (~5200 ha) of currently degraded area in Oregon. Despite the limited spatial extent, restoration-based NCS activities also provide important co-benefits [8], which warrant their inclusion in statewide conservation and climate strategies. Tidal wetland restoration provides a range of ecosystem services, including providing raw materials and food, maintaining

fisheries, and providing coastal protection and erosion control [163,164]. Similarly, restoration of sagebrush steppe from invasive annual grasses and avoided further conversion, which similarly contribute lower GHG reductions than other pathways, maintains habitat quality for a number of sagebrush-dependent species, including the Greater sage grouse (*Centrocercus urophasianus*), as well as limits the loss of other rangeland ecosystem services [111,165].

Other studies of climate mitigation on forest lands in the western United States have included wildfire mitigation and management activities [8,129,166]. Forest health treatments are critical for forest resilience and community safety. However, substantial uncertainties remain with respect to fire emission estimates and the timeframe for accrual of climate benefits from wildfire mitigation estimates. Climate benefits related to wildfire mitigation activities depend on the probability of a silvicultural treatment experiencing wildfire within the effective lifespan of the treatment, the difference in wildfire severity between treated and untreated alternatives, the level of emissions from a wildfire, and the cumulative impact of landscape scale interactions between forest fuels, treatment location, topography, climatic conditions and fire dynamics [167–170]. Silvicultural treatments aimed at reducing wildfire scope and severity result in immediate and short-term carbon emissions but can increase carbon storage and stability, particularly over many-decade long timeframes and when treatments are implemented across large spatial scales [171–173]. However, these benefits may not be realized within the timeframe of our study and may not accrue on an individual per-unit-area of treatment implementation basis.

In all of the scenarios used in this study, we assume that the implementation of NCS is ramped up over the next decade. For some activities, implementation continues to increase over the 30-year simulation period while other NCS implementation levels off after 2030. The actual contribution of NCS to GHG reduction goals will depend on the rate at which NCS increases across the landscape. Rather than predict the rates of NCS implementation based on socio-economic constraints, the scenarios we explore here offer answers to hypothetical "what if" questions about NCS implementation consistent with recommendations that climate mitigation efforts include engagement of the land sector in addition to fossil fuel mitigation [1].

## Conclusions

NCS provide climate benefits by either increasing carbon sequestration or reducing GHG emissions by changing land management activities. While the potential for Oregon's carbon-rich coastal and montane forests to contribute to climate mitigation has been discussed elsewhere [121,127,174], our study considers the GHG reduction potential across multiple natural and working land sectors, including forests, sagebrush-steppe, coastal wetlands, grasslands, and agriculture, and multiple NCS strategies. Importantly, our study illustrates that NCS can contribute meaningfully to state-level GHG reduction strategies. Our results suggest that increased investments in carbon sequestration or avoided emissions from the land sector can help states to their current GHG reduction goals and achieve GHG reductions nearer to near zero emissions by 2050.

## Supporting information

**S1 Fig. Area considered for deferred timber harvest scenarios.** Deferred timber harvest was applied to counties (shaded green) where less than 50% of the forests are considered at high risk for wildfire.
(DOCX)

**S2 Fig.** Multiple ownerships provide timber volume (A) and greenhouse gas emission reduction (B) under baseline and NCS implementation scenarios of timber harvest.
(DOCX)

**S1 Table. County designation across the interior to coastal productivity gradient.** These classifications were used to assign forest productivity rates, simplified to "west" and "east" regions to signify coastal versus interior sensu (1), for most forest-based NCS activities.
(DOCX)

**S2 Table. Average annual timber harvest in cubic meters (baseline) and percent deferment under each scenario by ownership.** Data are summarized from Oregon Department of Forestry harvest data (2000–2017) for the following counties: Benton, Clackamas, Clatsop, Columbia, Coos, Curry, Deschutes, Douglas, Klamath, Lane, Lake, Lincoln, Linn, Marion, Morrow, Multnomah, Polk, Tillamook, Washington, Yamhill.
(DOCX)

**S3 Table. Estimated annual reductions in MMTCO2e for each NCS activity under three different implementation scenarios in years 2035 and 2050.** Upper and lower bounds of 90% confidence interval are shown in parentheses.
(DOCX)

## Acknowledgments

Thank you to Dick Vander Schaaf and Allison Aldous for their advice and review of the tidal wetland methodology. We are also grateful to Renee Davis, Mark Stern, and Nick Sirovatka for their comments on methods and early versions of this research.

## Author Contributions

**Conceptualization:** Rose A. Graves, Ryan D. Haugo, Andrés Holz, Max Nielsen-Pincus, Cathy Macdonald, Kenneth Popper.

**Data curation:** Rose A. Graves, Bryce Kellogg, Michael Schindel.

**Formal analysis:** Rose A. Graves, Aaron Jones.

**Funding acquisition:** Ryan D. Haugo, Andrés Holz, Max Nielsen-Pincus.

**Investigation:** Rose A. Graves, Aaron Jones, Bryce Kellogg, Kenneth Popper, Michael Schindel.

**Methodology:** Rose A. Graves, Ryan D. Haugo, Bryce Kellogg, Kenneth Popper, Michael Schindel.

**Visualization:** Rose A. Graves.

**Writing – original draft:** Rose A. Graves.

**Writing – review & editing:** Rose A. Graves, Ryan D. Haugo, Andrés Holz, Max Nielsen-Pincus, Cathy Macdonald.

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
