## [Decision Letter · Decision Letter 0]

6 Sep 2019

PONE-D-19-20116

Potential greenhouse gas reductions from Natural Climate Solutions in Oregon, USA

PLOS ONE

Dear Dr. Graves,

Thank you for submitting your manuscript to PLOS ONE. After careful consideration, we feel that it has merit but does not fully meet PLOS ONE’s publication criteria as it currently stands. Therefore, we invite you to submit a revised version of the manuscript that addresses the points raised during the review process.

We would appreciate receiving your revised manuscript by Oct 21 2019 11:59PM. To enhance the reproducibility of your results, we recommend that if applicable you deposit your laboratory protocols in protocols.io, where a protocol can be assigned its own identifier (DOI) such that it can be cited independently in the future. For instructions see: http://journals.plos.org/plosone/s/submission-guidelines#loc-laboratory-protocols

We look forward to receiving your revised manuscript.

Kind regards,

Debjani Sihi

Academic Editor

PLOS ONE

Journal Requirements:

Additional Editor Comments (if provided):

Reviewers' comments:

Reviewer's Responses to Questions

**Comments to the Author**

1. Is the manuscript technically sound, and do the data support the conclusions?

Reviewer #1: No

Reviewer #2: Yes

2. Has the statistical analysis been performed appropriately and rigorously? 

Reviewer #1: Yes

Reviewer #2: N/A

3. Have the authors made all data underlying the findings in their manuscript fully available?

Reviewer #1: Yes

Reviewer #2: Yes

4. Is the manuscript presented in an intelligible fashion and written in standard English?

Reviewer #1: Yes

Reviewer #2: Yes

5. Review Comments to the Author

Reviewer #1: Graves et al., presented a well written and important manuscript describing the carbon mitigation potential of variaous Natural Climate Solutions (NCS). I believe this manuscript will be of interest to many scientists and policy makers examining mechanisms of carbon sequstration and mitigation. This is especially important as Oregon makes moves to regulate carbon emissions through cap and trade or other mechanisms. I have several suggestions to improve the manuscript, but my main criticism is that the coparisons between some of the NCSs are maybe not appropriate. The basis of some of hte NCSs are the entire landbase - timber harvesting for example is based on all land across all ownerships. Whereas, riparian restoration is only based on the current rate of restorration. The authors point out that their calculated rate of riparian restoration may be an underestimate because they onyl use the rate of restortion reported by a couple afgencies. However, this rate is probably huge underestimate given the length of streams and rivers that pass thorugh agricultural and urban landscapes in need of restoration (development or harvesting up to stream edge). At the very least, I would like to see the authors make an attempt at determining the amount of C that could be sequestered in riparian zone if they were all restored - you could use OFPA to determine the width of riparian zones by stream type (interesting that once managed forest land is converted to urban or ag the riparian zones can be almost non-exisent, I digress). As it stands, the basis for each of these NCSs is different and so tough to compare.

Specific comments.

Line 112: The authors shoudl consider creating a conceptual diagram or figure that describes their methods.

Table 1: Consider adding citations to this table

Line 175-177: Was this arbitrarily chosen? No problem if so, but if guided by some research it needs a citation.

Reviewer #2: Potential greenhouse gas reductions from Natural Climate Solutions in Oregon, USA

Graves, Haugo, Holz, Nielsen-Pincus, Joens, Kellogg, MacDonald, Popper

Overview

This study sought to determine the relative importance of different natural climate solutions (NCS) on their ability to reduce greenhouse gas emissions. The study was comprehensive, covering many different options for NCS, and the results were interesting, showing that delayed timber harvest was the most important contributor to potential GHG reductions. The paper was well-written, figures were generally clear and the results had state-level policy implications, which was good to see. My main concern was the lack of information in the discussion about the potential importance of different landowners in delaying timber harvest to reduce GHG emissions. If it was mostly private industrial, it would be a lot less practical to implement than say state lands.

Minor editorial comments are listed below.

Abstract

L31 Clarify “avoided conversion”. Do you mean to non-forest and to ag? I don’t think natural and working land(s) is necessary.

L33. “the” global drawdown

L35. Plural? Natural and working lands.

L40. Why only on federal lands? Above it makes it sound like you’re looking across all ownerships

L43. Is avoided conversion the highest on a per area basis? You might want to say that rather than saying it’s relatively high.

L51. It’s true that it’s dependent on coordination across jurisdictions but you didn’t discuss private industrial or non-industrial shifts so it’s not as relevant here. Somewhere here you need to address the scope of your study in terms of ownership so it’s clear.

L53. I would just say state level.

Introduction

L60. Avoid “the worst effects of climate change” sounds better.

L61. Again lands?

L67. “The next” 10 to 15 years.

L74. How about just “some states”? I’m not sure it matters if they are “influential” are not, in terms of emissions.

L78. First-of-its-kind

L81. You repeat “natural and working lands” fairly frequently. This might be a place where NCS belongs anyway.

L87. Comma before including would be helpful.

L88. To develop a carbon policy framework seems unnecessary. Or at least makes the sentence too long and confusing.

L89. Land management rather than natural and working lands?

L90. Comma before promoting.

L94. Comma after lead.

L109. Omit “given different…”. I think it detracts from the power of the sentence.

Methods

Methods are quite long at 19 pages. Maybe you could shorten the introductory paragraph to each type of NCS.

L71. Omit “for testing”

Table 1. The text on the left is shifted too far to the right (i.e. Avoided Conversion). Switch to single spacing there?

Under nutrient management, it might be better to say “avoided emissions by…”

Interesting because salvage logging and site prep is common practice after wildfire in OR. I wonder why you omitted it. Maybe you’ll tell me later.

Figure 1. It’s not exactly clear why you separated ages of replanting. Is it because of stand age? We don’t know when the fires will occur. Or are you trying to stagger it? Anyway, it’s not clear why you analyze years differently here, but not with the other treatments.

L195. I wasn’t clear if you included the emissions from the wildfires.

L210. Did you assume the avoided emission between aboveground biomass was similar between grasses and crops?

L219. Should be carbon-rich

L219. Add comma after world.

L222. Omit “in-forest” throughout the paper. Or just put forest.

L231. Interesting. So this over-estimates the emissions now, but they will occur after the simulation.

L242. Belowground is one word.

L275. These figures seem like they belong in the results.

L286. You say it could be used on most of the ag acres, but you haven’t gotten to the point where you tell us how much its being used already. How about Cover crops can be used in Oregon…”?

L307. You don’t tell us whey no-till is important, like you did for cover crops. And you put national figures here but not in the cover crops. It would be good to be consistent in the text across treatments.

L364. I don’t understand why you limited it to fed lands when everyone usually replant. I think you just need to clarify your justification. Are you assuming that OFPA could be revoked on federal but not primate land? What about state forested land?

Results

Figure 1. What about color coding it like you did Figure 3?

Figure 2. You focus on 2035, but it’s not marked on your graph. Maybe add a vertical dashed gray line?

L510. Is the median range you present across all 3 scenarios? I feel like I wanted one value with a SE/SD? Or 3 medians, one for each scenario.

Figure 5 resolution is poor.

L550. I’m not sure how it could be as high as 67%. Please clarify.

Discussion

L563. I’m not sure I would consider a max of 20% as substantial.

L596. I was curious to know which landowners provided the most emissions reductions since it was the most important factor. Could you add this analysis to the paper? Also consider adding a paragraph about landowners and their expected flexibility to altering their management practices to defer harvest.

Conclusion

L3667. But changing would include limiting or increasing. Might want to clarify or remove if not necessary.

6. PLOS authors have the option to publish the peer review history of their article (what does this mean?). If published, this will include your full peer review and any attached files.

Reviewer #1: No

Reviewer #2: No

---

## [Author Response · Author response to Decision Letter 0]

9 Nov 2019

We are writing to resubmit our manuscript, “Potential greenhouse gas reduction from Natural Climate Solutions in Oregon, USA”, which has been revised to reflect the comments from two reviewers. I apologize for the added time it has taken to respond to these reviews and appreciate your extending the deadline to accommodate my maternity leave. We modified our manuscript substantially in response to the reviewers’ comments and thank them for their feedback. Specifically, we have addressed the major concerns of Reviewer 1 by updating our estimates of riparian reforestation potential and including this estimate in our scenarios of carbon sequestration. We address Reviewer 2’s major concern by adding an analysis of timber harvest and avoided emissions from timber harvest deferral attributed to different ownerships, as well as a discussion of the possible challenges associated with those ownerships. Furthermore, we have reorganized and streamlined our methods section and have made every attempt to address the reviewers’ remarks. 

We provide our response to specific comments from the reviewers below. Our response is in italics and we include line number references to the revised manuscript where appropriate. 

Thank you for your consideration. We look forward to hearing from you in due course.

Sincerely, 

Rose A. Graves

Detailed Response to Reviewer Comments

Authors’ response to reviewer comments follow the initial reviewer comment. 

Reviewer #1: Graves et al., presented a well written and important manuscript describing the carbon mitigation potential of variaous Natural Climate Solutions (NCS). I believe this manuscript will be of interest to many scientists and policy makers examining mechanisms of carbon sequstration and mitigation. This is especially important as Oregon makes moves to regulate carbon emissions through cap and trade or other mechanisms. 

Thank you for your positive comments on the utility of our research. 

I have several suggestions to improve the manuscript, but my main criticism is that the coparisons between some of the NCSs are maybe not appropriate. The basis of some of hte NCSs are the entire landbase - timber harvesting for example is based on all land across all ownerships. Whereas, riparian restoration is only based on the current rate of restorration. The authors point out that their calculated rate of riparian restoration may be an underestimate because they onyl use the rate of restortion reported by a couple agencies. However, this rate is probably huge underestimate given the length of streams and rivers that pass thorugh agricultural and urban landscapes in need of restoration (development or harvesting up to stream edge). At the very least, I would like to see the authors make an attempt at determining the amount of C that could be sequestered in riparian zone if they were all restored - you could use OFPA to determine the width of riparian zones by stream type (interesting that once managed forest land is converted to urban or ag the riparian zones can be almost non-exisent, I digress). As it stands, the basis for each of these NCSs is different and so tough to compare.

We respectfully disagree with the Reviewer’s assertion that comparisons across NCS activities are not appropriate. In all cases, increased NCS implementation in the Limited, Moderate, and Ambitious Scenarios is based on a change from the baseline rate of an activity. In some cases, an activity occurs across multiple ownerships and is reported based on those ownerships (i.e., timber harvest). In other cases, reported baseline rates of an activity are not tied to ownership categories (i.e., riparian reforestation). We used a similar approach for all of the NCS activities, regardless of whether they applied to all ownerships or only a specific portion of the land base. Furthermore, the comparisons presented among NCS in our study are similar to published comparisons of GHG reduction activities due to NCS (Cameron et al. 2017, Griscom et al. 2017, Fargione et al. 2018). As with these published estimates, we do not claim to provide a complete inventory and assessment of all possible pathways but rather to highlight and start a conversation around the role of NCS in climate change mitigation. 

We have addressed the Reviewer’s concern with respect to the underestimate of riparian area restoration. We maintain our current methodology for establishing the baseline annual rate of restoration [see lines 380-387] as the best first approximation based on the data available. No current data products, spatial or otherwise, exist which accurately identify statewide riparian restoration efforts. We rely on reporting to two of the largest riparian restoration funding agencies within the State (Oregon Watershed Enhancement Board and the Natural Resource Conservation Service). Based on these data, we estimate that the current rate of riparian reforestation (i.e., restoration plantings) is 2395 ha per year, with an average of 1712 ha per year replanted east of the Cascades and 683 ha per year replanted west of the Cascades. 

To address the Reviewer’s concern, we have improved on the upper bounds of riparian reforestation potential used in the Ambitious Scenario estimate of carbon sequestration potential. While detailed maps of current riparian condition exist for individual sections of selected watersheds throughout Oregon, a comprehensive statewide assessment of riparian condition has not been conducted for the State of Oregon. The method offered by Reviewer #1 using the Oregon Forest Practices Act (OFPA) stream designations is not sufficient for estimating restorable riparian areas for two main reasons. First, the OFPA buffer requirements vary depending on stream characteristics from 20 to 100 m and are not easily translated to mapped buffers at the state level. Second, while buffering all the streams in the state may provide an estimate of riparian habitat, it still does not adequately describe the current condition of those riparian areas. For example, an oversimplified estimate of statewide riparian areas using 100-m buffers on either side of mapped streams would result in a total area of riparian habitat in Oregon of 36,927 km2 but would not take into consideration stream order/size or condition and is likely to overestimate of potential restoration opportunity (Gregory 2000). 

Instead, we estimate the maximum extent of riparian reforestation opportunity by combining published floodplain maps (Wing et al. 2018) with recent mapped tree canopy cover data (USFS 2019) and environmental site potential (LANDFIRE 2014). We considered areas to have riparian reforestation potential if they (1) occurred within a 100-year floodplain, (2) had less than 40% canopy cover, and (3) had ecological site potential that included forest, woodland, or was undetermined based on biophysical setting. Urban areas are outside the scope of this study and so we did not include areas mapped as high or moderate density development (NLCD 2011) in our estimate of riparian reforestation potential. This resulted in 202,415 ha (2024 km2) of potential reforestation across Oregon, the majority of which is located on the west side of the state. We have modified our Ambitious Scenario to reach a cumulative riparian reforestation of 202,415 ha by 2050 [Table 3]. We also included the maximum threshold in all three scenarios, better reflecting the upper limit of riparian reforestation opportunity in the eastern portion of the state. These changes led to increased C sequestration potential from riparian reforestation as compared to our previous estimate in both the Moderate and Ambitious scenarios. Specifically, by 2050, the cumulative C sequestration potential from riparian reforestation increased by 11% or 3.8 MMT CO2e in the Ambitious Scenario and by 86% or 5.1 MMT CO2e in the Moderate Scenario. 

Specific comments.

Line 112: The authors should consider creating a conceptual diagram or figure that describes their methods.

Thank you for the suggestion. Rather than create an additional figure or conceptual diagram, we have elected to streamline and simplify the “General Analytical Framework” section and hope that it helps to illustrate the cross-cutting methods. 

Table 1: Consider adding citations to this table

We draw the definitions of NCS pathways from Griscom et al. (2017), Cameron et al. (2017), and Fargione et al. (2018). We have added these citations to the table caption. 

Line 175-177: Was this arbitrarily chosen? No problem if so, but if guided by some research it needs a citation.

We have added citations and have revised our estimate of the proportion of carbon sequestration lost due to the conversion of forests to urban and rural land uses [lines 164-166]. Woodbury and colleagues (2007) assessed carbon flux across the United States and estimate that urban/suburban forests sequester carbon at a ratio of 16/100 compared to forests. Thus, we revised our estimates of forgone C sequestration in forests converted to urban land use to be 84% of the current sequestration. For rural land uses, we assume that 50% of the forest cover is retained, loosely following the NLCD low-intensity development definition (<49% impervious cover). 

Reviewer #2: Potential greenhouse gas reductions from Natural Climate Solutions in Oregon, USA

Graves, Haugo, Holz, Nielsen-Pincus, Joens, Kellogg, MacDonald, Popper

Overview

This study sought to determine the relative importance of different natural climate solutions (NCS) on their ability to reduce greenhouse gas emissions. The study was comprehensive, covering many different options for NCS, and the results were interesting, showing that delayed timber harvest was the most important contributor to potential GHG reductions. The paper was well-written, figures were generally clear and the results had state-level policy implications, which was good to see. My main concern was the lack of information in the discussion about the potential importance of different landowners in delaying timber harvest to reduce GHG emissions. If it was mostly private industrial, it would be a lot less practical to implement than say state lands.

Thank you for your positive review of our manuscript. We appreciate your suggestions and address the specific comments below. With respect to the delayed timber harvest pathway, we have followed your suggestion and added results specifically outlining delayed timber harvest by ownership [see lines 489-497] as well as a discussion of the potential implications the different ownerships have for implementation [see lines 560-584]. 

Minor editorial comments are listed below.

Abstract

L31 Clarify “avoided conversion”. Do you mean to non-forest and to ag? I don’t think natural and working land(s) is necessary.

L33. “the” global drawdown

L35. Plural? Natural and working lands.

We have made the changes suggested above [lines 31 – 35].

L40. Why only on federal lands? Above it makes it sound like you’re looking across all ownerships

Each NCS activity has a specific available land base – we only consider additional replanting after wildfire on Federal lands given current requirements for replanting on private lands. We have removed the reference to federal lands in the abstract to avoid confusion and have added a sentence to clarify that each NCS was evaluated based on practice-specific assumptions and applicable land base [line 37].

L43. Is avoided conversion the highest on a per area basis? You might want to say that rather than saying it’s relatively high.

Avoided conversion of forests does not have the highest estimated per area GHG reduction (MTCO2e ha-1 yr-1). It is less than riparian reforestation and tidal wetland restoration (see Figure 1). We have not made any changes to the wording here. 

L51. It’s true that it’s dependent on coordination across jurisdictions but you didn’t discuss private industrial or non-industrial shifts so it’s not as relevant here. Somewhere here you need to address the scope of your study in terms of ownership so it’s clear.

We have removed the reference to jurisdictions and have added a sentence to the abstract to clarify the scope of our study in terms of ownership [lines 37 - 39].

L53. I would just say state level. We have made the suggested change.

Introduction

L60. Avoid “the worst effects of climate change” sounds better. We have made the suggested change.

L61. Again lands? We have made the suggested change.

L67. “The next” 10 to 15 years. We have made the suggested change.

L74. How about just “some states”? I’m not sure it matters if they are “influential” are not, in terms of emissions. We have made the suggested change.

L78. First-of-its-kind We have made the suggested change.

L81. You repeat “natural and working lands” fairly frequently. This might be a place where NCS belongs anyway. Thank you for pointing out our repetitive wording. We have made the suggested change.

L87. Comma before including would be helpful. We have made the suggested change.

L88. To develop a carbon policy framework seems unnecessary. Or at least makes the sentence too long and confusing. We have simplified the wording. 

L89. Land management rather than natural and working lands? We have made the suggested change.

L90. Comma before promoting. We have made the suggested change.

L94. Comma after lead. We have made the suggested change.

L109. Omit “given different…”. I think it detracts from the power of the sentence. We have made the suggested change.

Methods

Methods are quite long at 19 pages. Maybe you could shorten the introductory paragraph to each type of NCS.

We have attempted to streamline the methods section but still retain enough detail to adequately describe our process. Specifically, we have shortened the cross-cutting methods [lines 113 – 129], removed the subheadings under each activity, and shortened the introductory paragraphs for each NCS activity, as suggested by the Reviewer. 

L71. Omit “for testing” We have made the suggested change.

Table 1. The text on the left is shifted too far to the right (i.e. Avoided Conversion). Switch to single spacing there? We have made the suggested change.

Under nutrient management, it might be better to say “avoided emissions by…” We have made the suggested change.

Interesting because salvage logging and site prep is common practice after wildfire in OR. I wonder why you omitted it. Maybe you’ll tell me later.

Figure 1. It’s not exactly clear why you separated ages of replanting. Is it because of stand age? We don’t know when the fires will occur. Or are you trying to stagger it? Anyway, it’s not clear why you analyze years differently here, but not with the other treatments.

The Reviewer is correct that we do not know when fires will occur. We make the assumption that the disturbance pattern of wildfires (i.e., frequency and extent) will not change from the historical pattern. We simulate fire frequency and extent based off historical trends [lines 350 – 355]. Replanting after wildfire is a single activity, but has varying rates of sequestration compared to natural regeneration after wildfire depending on stand age [lines 342 - 349]. Thus, we include the varying sequestration rates within Figure 1. 

L195. I wasn’t clear if you included the emissions from the wildfires.

We do not include emissions from the wildfires per se. The estimates of lost biomass and carbon sequestration are based on comparisons of intact sagebrush-steppe communities to invasive annual grass dominated communities. 

L210. Did you assume the avoided emission between aboveground biomass was similar between grasses and crops?

Following Fargione et al. 2018, we assume that annual aboveground biomass loss is similar in grasslands and croplands. In both, aboveground biomass is annually harvest, burned, grazed, or decomposed within a few years and thus does not contribute to avoided emissions. 

L219. Should be carbon-rich We have made the suggested change, and have moved this sentence to the discussion. 

L219. Add comma after world. We respectfully disagree, as there are not two independent clauses in this sentence.

L222. Omit “in-forest” throughout the paper. Or just put forest. We have made the suggested change.

L231. Interesting. So this over-estimates the emissions now, but they will occur after the simulation. The Reviewer is correct. At the end of the 30-years, if harvest levels return to current levels the emissions will return to current levels but forest carbon stocks will be higher [as stated Line 222-226]. 

L242. Belowground is one word. We have changed this throughout the manuscript. 

L275. These figures seem like they belong in the results. To remain consistent in our organization across NCS activities, we have retained the numbers referenced here (i.e., the per unit mitigation potential). The estimates of per unit mitigation potential from NCS activities are intermediate methods and are not considered main results in our study. 

L286. You say it could be used on most of the ag acres, but you haven’t gotten to the point where you tell us how much its being used already. How about Cover crops can be used in Oregon…”? We have made the suggested change to this sentence, and also have moved this sentence to the discussion. 

L307. You don’t tell us why no-till is important, like you did for cover crops. And you put national figures here but not in the cover crops. It would be good to be consistent in the text across treatments. We have changed the organization of our Methods section and moved this text (along with other introductory text for NCS activities) to the introduction or discussions sections. 

L364. I don’t understand why you limited it to fed lands when everyone usually replant. I think you just need to clarify your justification. Are you assuming that OFPA could be revoked on federal but not primate land? What about state forested land?

As the Reviewer indicates, we assume that lands covered by the Oregon Forest Practices Act (i.e., private and state lands) generally are salvage harvested after wildfire, thus triggering the requirement to replant. Therefore, we assume that there is little additional C sequestration opportunity above the business-as-usual management practice. Conversely, current practice on federal lands does not include such high rates of replanting after wildfire [see our estimate of replanting rates on Federal land, Table 2] and thus represents an opportunity for increased activity above the business-as-usual management practice. 

Results

Figure 1. What about color coding it like you did Figure 3? We have made the suggested change.

Figure 2. You focus on 2035, but it’s not marked on your graph. Maybe add a vertical dashed gray line? We have added 2035 to the x-axis to make this graph easier to read. 

L510. Is the median range you present across all 3 scenarios? I feel like I wanted one value with a SE/SD? Or 3 medians, one for each scenario. We have modified our results to provide the median along with upper and lower bounds for each scenario.

Figure 5 resolution is poor. We have improved the resolution of this figure. 

L550. I’m not sure how it could be as high as 67%. Please clarify. By 2050, if the State is successful in meeting its current fossil fuel mitigation goals across other sectors (e.g., transportation, energy, housing) then annual GHG emissions will be close to 14 MMTCO2e. Our Ambitious Scenario suggests that increased NCS implementation could mitigate an additional 9.74 MMTCO2e per year, which is 69% of the remaining GHG emissions in 2050. 

We have updated this section for clarity and also to include new results.

Discussion

L563. I’m not sure I would consider a max of 20% as substantial. We have changed the wording. 

L596. I was curious to know which landowners provided the most emissions reductions since it was the most important factor. Could you add this analysis to the paper? Also consider adding a paragraph about landowners and their expected flexibility to altering their management practices to defer harvest. 

We have added two figures, in addition to the pre-existing S2 Table, which describe the relative contribution of different ownerships to both the baseline timber harvest as well as the cumulative GHG reductions from deferred harvest in the NCS Scenarios [S2 Fig A & B] and describe this breakdown in our results [lines 486 - 494]. Finally, we have added more of a discussion regarding the different landowners and their potential flexibility/ability to alter their management practices [lines 560 – 584].

Conclusion

L3667. But changing would include limiting or increasing. Might want to clarify or remove if not necessary. We changed the wording here.

---

## [Decision Letter · Decision Letter 1]

23 Jan 2020

PONE-D-19-20116R1

Potential greenhouse gas reductions from Natural Climate Solutions in Oregon, USA

PLOS ONE

Dear Dr. Graves,

Thank you for submitting your manuscript to PLOS ONE. After careful consideration, we feel that it has merit but does not fully meet PLOS ONE’s publication criteria as it currently stands. Therefore, we invite you to submit a revised version of the manuscript that addresses the points raised during the review process.

We would appreciate receiving your revised manuscript by Mar 08 2020 11:59PM. To enhance the reproducibility of your results, we recommend that if applicable you deposit your laboratory protocols in protocols.io, where a protocol can be assigned its own identifier (DOI) such that it can be cited independently in the future. For instructions see: http://journals.plos.org/plosone/s/submission-guidelines#loc-laboratory-protocols

We look forward to receiving your revised manuscript.

Kind regards,

Debjani Sihi

Academic Editor

PLOS ONE

Reviewers' comments:

Reviewer's Responses to Questions

**Comments to the Author**

1. If the authors have adequately addressed your comments raised in a previous round of review and you feel that this manuscript is now acceptable for publication, you may indicate that here to bypass the “Comments to the Author” section, enter your conflict of interest statement in the “Confidential to Editor” section, and submit your "Accept" recommendation.

Reviewer #2: All comments have been addressed

Reviewer #3: All comments have been addressed

Reviewer #4: (No Response)

2. Is the manuscript technically sound, and do the data support the conclusions?

Reviewer #2: Yes

Reviewer #3: Yes

Reviewer #4: Yes

3. Has the statistical analysis been performed appropriately and rigorously? 

Reviewer #2: N/A

Reviewer #3: No

Reviewer #4: Yes

4. Have the authors made all data underlying the findings in their manuscript fully available?

Reviewer #2: No

Reviewer #3: Yes

Reviewer #4: Yes

5. Is the manuscript presented in an intelligible fashion and written in standard English?

Reviewer #2: Yes

Reviewer #3: Yes

Reviewer #4: Yes

6. Review Comments to the Author

Reviewer #2: The authors have addressed all my comments to my satisfaction. Overall, the manuscript looks good and is publishable. My only concern was about data availability. The authors state that all the data is available in the paper and supplemental, but I couldn't find any numbers etc that would make the work reproducible. Though maybe I missed it.

Reviewer #3: This is an important study summarizing the NCS-effects of conservation, management, and restoration on the land base in Oregon. A strength of the paper is the consideration of various alternative management strategies for many types of land uses.

While it may not be specifically an NCS, one emerging use of timber that has a large (potential) GHG reduction effect is the substitution of cross-laminated mass timber for concrete and steel. Concrete and steel create large CO2 emissions. Fain et al. (2018) is not cited. They found that substitution is a key variable when assessing carbon benefits over time. I do not know if such substitution effects fall into the scope of NCS, but regardless it appears to be significant...Fain et al claim for over 100 years, substituting wood for concrete and steel has more carbon benefits than lengthening rotations to 120 years. Similarly, wood pellets substituting for natural gas reduce leakage from the wood products pipeline. Such wood products may decrease the NCS carbon benefit but could have non-negligible contribution to fossil-fuel reduction shown in Fig. 5.

Fain et al, and many other studies, have assessments of different rotation lengths on landscape carbon storage, which would be worth comparing with your results. The numbers are difficult to compare as presented in each study but should be possible to do. The paragraph on lines 548-559 would be a good location to compare the results of this study and those of other studies (for example: Fain, Buotte, Franklin, Harmon, others). A comparison of the quantitative results from these previous studies that consider longer rotations, or partial harvests, should be expected from readers. Such a comparison, and ideally an assessment of which study is more robust, would be very valuable.

Fain, S., Kittler, B., & Chowyuk, A. (2018). Managing Moist Forests of the Pacific Northwest United States for Climate Positive Outcomes. Forests, 9(10), 618. doi: 10.3390/f9100618

Buotte, P. C., Law, B. E., Ripple, W. J., & Berner, L. T. (2019). Carbon sequestration and biodiversity co‐benefits of preserving forests in the western United States. Ecological Applications. doi: 10.1002/eap.2039

170-174: I do not understand how using the SD of county data is an estimate of uncertainty in C emmissions. There is no expected central tendency in mean C numbers across space. I would expect large regional differences based on topography, rainfall, etc. even within west-side and east-side values. I think that uncertainty should be based upon uncertainty in the original estimates of AGB...which would require examining the methods behind obtaining these estimates. On the other hand, if you can make the argument that county-level estimates for Douglas-fir forest in, say, Clackamas County vs. Josephine County should be similar...then state that is the case.

193 and 202: random normal distributions? or randomly from a normal distribution? Also, normal distributions have a mean and sd by definition, so do not need to specify that, unless you want to provide the mean and sd values in the text (or at least point to the table that has these data.) Similarly, line 288 and 310...'random?' uniform distribution?

Minor comments

52: low potential of...

Reviewer #4: I was not one of the previous reviewers for this manuscript. After reading the reviews, the authors’ responses, and the entire paper, I suggest that it be accepted for publication after minor revision. This is a well-done study that will have substantial interest in examining the benefits of NCS in a single state. It is important that global and U.S.-level NCS studies be regionalized to determine the importance of local factors in the potential outcomes. Oregon’s strong land-use laws indeed have a strong impact on what can achieved in NCS because many land-use conversions are already not allowed. All NCS studies are fraught with assumptions based upon imperfect data and uncertainties on what is feasible politically and economically, but the authors make reasonable decisions and clearly state the limitations of their study.

I have a few, mostly minor, suggestions for the authors:

Table 2. MBF should be defined.

Table S2. For average annual harvest, use commas to delineate changes at each 1,000 change and right justify the numbers so that similar digits (i.e., in 10s) line up vertically. Currently, it is very difficult to visually compare the areas.

Line 388. Given the uncertainties in these estimates, round up these rates to the 1's place (i.e., get rid of the numbers to the right of the decimal).

Line 652. This is my only substantive criticism. It is good that the Law et al. (2018) article in PNAS is mentioned here, but I think that its findings should be elaborated on more in the discussion. It examines many of the forestry activities in Oregon that are shown to have the highest NCS benefit in this manuscript but with substantially different methods and assumptions. Thus, I believe that it would enhance the discussion substantially to compare the results of the two studies.

7. PLOS authors have the option to publish the peer review history of their article (what does this mean?). If published, this will include your full peer review and any attached files.

Reviewer #2: No

Reviewer #3: No

Reviewer #4: No

---

## [Author Response · Author response to Decision Letter 1]

29 Jan 2020

Response to Reviewer Comments

Authors’ response to reviewer comments are in blue and italics. 

Review Comments to the Author

Reviewer #2: The authors have addressed all my comments to my satisfaction. Overall, the manuscript looks good and is publishable. My only concern was about data availability. The authors state that all the data is available in the paper and supplemental, but I couldn't find any numbers etc that would make the work reproducible. Though maybe I missed it.

Thank you for your comments. In order to make the data availability more transparent and the work reproducible, we will archive the R code used in our simulations as well as the input files on GitHub before publication of the study. 

Reviewer #3: This is an important study summarizing the NCS-effects of conservation, management, and restoration on the land base in Oregon. A strength of the paper is the consideration of various alternative management strategies for many types of land uses.

Thank you for your positive comments. 

While it may not be specifically an NCS, one emerging use of timber that has a large (potential) GHG reduction effect is the substitution of cross-laminated mass timber for concrete and steel. Concrete and steel create large CO2 emissions. Fain et al. (2018) is not cited. They found that substitution is a key variable when assessing carbon benefits over time. I do not know if such substitution effects fall into the scope of NCS, but regardless it appears to be significant...Fain et al claim for over 100 years, substituting wood for concrete and steel has more carbon benefits than lengthening rotations to 120 years. Similarly, wood pellets substituting for natural gas reduce leakage from the wood products pipeline. Such wood products may decrease the NCS carbon benefit but could have non-negligible contribution to fossil-fuel reduction shown in Fig. 5.

Fain et al, and many other studies, have assessments of different rotation lengths on landscape carbon storage, which would be worth comparing with your results. The numbers are difficult to compare as presented in each study but should be possible to do. The paragraph on lines 548-559 would be a good location to compare the results of this study and those of other studies (for example: Fain, Buotte, Franklin, Harmon, others). A comparison of the quantitative results from these previous studies that consider longer rotations, or partial harvests, should be expected from readers. Such a comparison, and ideally an assessment of which study is more robust, would be very valuable.

Thank you for the references. We agree that substitution is an important, albeit uncertain, variable in calculating the carbon emissions from timber harvest. We do not include changes in the current wood products pool in our NCS scenarios but have now added a short discussion of the importance of long-lived wood products (such as cross-laminate timber), including references to Fain et al. 2018 [Lines 576 – 588].

Of the harvested wood volume considered in our study, 15% is assumed to be unused mill residue or mill residue burned on site. We consider the remaining 85% to be the transformed wood products pool. We further assume that 72% of the transformed wood products in Oregon are long-term storage (i.e., the carbon remains stored in these products for 20 years or more), following Fargione et al. 2018 and Oswalt et al. 2018. While our study considers deferring overall timber harvest as the mechanism to reduce timber harvest emissions, increasing the proportion of transformed wood products in the long-term storage pool would also result in lower timber harvest emissions regardless of the substitution effect. 

Fain, S., Kittler, B., & Chowyuk, A. (2018). Managing Moist Forests of the Pacific Northwest United States for Climate Positive Outcomes. Forests, 9(10), 618. doi: 10.3390/f9100618

Buotte, P. C., Law, B. E., Ripple, W. J., & Berner, L. T. (2019). Carbon sequestration and biodiversity co‐benefits of preserving forests in the western United States. Ecological Applications. doi: 10.1002/eap.2039

Thank you for the suggested reference to Buotte et al. 2019. This paper was not published at the time of our submission. We now include it, along with others you suggested, in the discussion (Lines 572-573). 

170-174: I do not understand how using the SD of county data is an estimate of uncertainty in C emmissions. There is no expected central tendency in mean C numbers across space. I would expect large regional differences based on topography, rainfall, etc. even within west-side and east-side values. I think that uncertainty should be based upon uncertainty in the original estimates of AGB...which would require examining the methods behind obtaining these estimates. On the other hand, if you can make the argument that county-level estimates for Douglas-fir forest in, say, Clackamas County vs. Josephine County should be similar...then state that is the case.

We thank the Reviewer for pointing out this shortcoming in our estimate of uncertainty for avoided emissions and lost sequestration potential due to conversion of forests to development. We have updated our methods (see Lines 152-159 and Lines 173 - 177). Specifically, we now take advantage of the ability to report the sampling error within the FIA Evalidator tool. This allowed us to get estimates of the mean carbon in all forest carbon pools (aboveground, belowground, woody debris, litter, and soil organic carbon) as well as the sampling error and the number of plots used for those calculations in each county. We retain our groupings of counties into interior and coastal regions (east side vs. west side) but rather than calculate the standard deviation across the counties, we pool data within each region and calculate a pooled mean and standard deviation for the interior and coastal region. We believe that this provides an adequate estimation of uncertainty for carbon emissions and sequestration in each region. 

193 and 202: random normal distributions? or randomly from a normal distribution? Also, normal distributions have a mean and sd by definition, so do not need to specify that, unless you want to provide the mean and sd values in the text (or at least point to the table that has these data.) Similarly, line 288 and 310...'random?' uniform distribution?

We have clarified the wording about the distributions used in Monte Carlo simulations throughout our manuscript (including lines 193, 202, 288, 310). 

Minor comments

52: low potential of...

Reviewer #4: I was not one of the previous reviewers for this manuscript. After reading the reviews, the authors’ responses, and the entire paper, I suggest that it be accepted for publication after minor revision. This is a well-done study that will have substantial interest in examining the benefits of NCS in a single state. It is important that global and U.S.-level NCS studies be regionalized to determine the importance of local factors in the potential outcomes. Oregon’s strong land-use laws indeed have a strong impact on what can achieved in NCS because many land-use conversions are already not allowed. All NCS studies are fraught with assumptions based upon imperfect data and uncertainties on what is feasible politically and economically, but the authors make reasonable decisions and clearly state the limitations of their study.

Thank you for your positive comments and assessment of our study. 

I have a few, mostly minor, suggestions for the authors:

Table 2. MBF should be defined.

Table S2. For average annual harvest, use commas to delineate changes at each 1,000 change and right justify the numbers so that similar digits (i.e., in 10s) line up vertically. Currently, it is very difficult to visually compare the areas.

Line 388. Given the uncertainties in these estimates, round up these rates to the 1's place (i.e., get rid of the numbers to the right of the decimal).

We have made the suggested changes above (Table 2, Table S2, and Line 388[400 in revised manuscript]). 

Line 652. This is my only substantive criticism. It is good that the Law et al. (2018) article in PNAS is mentioned here, but I think that its findings should be elaborated on more in the discussion. It examines many of the forestry activities in Oregon that are shown to have the highest NCS benefit in this manuscript but with substantially different methods and assumptions. Thus, I believe that it would enhance the discussion substantially to compare the results of the two studies.

We have added a more thorough comparison of our study with the work of Law et al. (2018) [Lines 567-575]. While a direct comparison of their work with our work is challenging given the different assumptions and accounting metric (i.e., Law and colleagues track the net ecosystem carbon balance whereas we focus solely on avoided emissions or increased sequestration due to changes in land management), our study concurs that limiting timber harvest provides the greatest GHG emission reductions.

---

## [Editor Report · Decision Letter 2]

2 Mar 2020

Potential greenhouse gas reductions from Natural Climate Solutions in Oregon, USA

PONE-D-19-20116R2

Dear Dr. Graves,

We are pleased to inform you that your manuscript has been judged scientifically suitable for publication and will be formally accepted for publication once it complies with all outstanding technical requirements.

With kind regards,

Debjani Sihi

Academic Editor

PLOS ONE
---

## [Editor Report · Acceptance letter]

20 Mar 2020

PONE-D-19-20116R2 

Potential greenhouse gas reductions from Natural Climate Solutions in Oregon, USA 

Dear Dr. Graves:

I am pleased to inform you that your manuscript has been deemed suitable for publication in PLOS ONE. Congratulations! Your manuscript is now with our production department. 

With kind regards,

on behalf of

Dr. Debjani Sihi 

Academic Editor

PLOS ONE